# Globally Convergent Variational Inference

**Declan McNamara**  **Jackson Loper**  **Jeffrey Regier**
Department of Statistics
University of Michigan
{declan, jaloper, regier}@umich.edu

## Abstract

In variational inference (VI), an approximation of the posterior distribution is selected from a family of distributions through numerical optimization. With the most common variational objective function, known as the evidence lower bound (ELBO), only convergence to a *local* optimum can be guaranteed. In this work, we instead establish the *global* convergence of a particular VI method. This VI method, which may be considered an instance of neural posterior estimation (NPE), minimizes an expectation of the inclusive (forward) KL divergence to fit a variational distribution that is parameterized by a neural network. Our convergence result relies on the neural tangent kernel (NTK) to characterize the gradient dynamics that arise from considering the variational objective in function space. In the asymptotic regime of a fixed, positive-definite neural tangent kernel, we establish conditions under which the variational objective admits a unique solution in a reproducing kernel Hilbert space (RKHS). Then, we show that the gradient descent dynamics in function space converge to this unique function. In ablation studies and practical problems, we demonstrate that our results explain the behavior of NPE in non-asymptotic finite-neuron settings, and show that NPE outperforms ELBO-based optimization, which often converges to shallow local optima.

## 1 Introduction

In variational inference (VI), the parameters $\eta$ of an approximation to the posterior $Q(\Theta; \eta)$ are selected to optimize an objective function, typically the evidence lower bound (ELBO) (Blei et al., 2017). However, the ELBO is generally nonconvex in $\eta$, even for simple variational families such as the family of Gaussian distributions, and so only convergence to a local optimum of the ELBO can be guaranteed (Ghadimi and Lan, 2015; Ranganath et al., 2014; Hoffman et al., 2013). As the number of such optima and the degree of suboptimality of each are generally unknown, the lack of global convergence guarantees constitutes a significant complication for practitioners and a longstanding barrier to the broader adoption of VI.

In this work, we present the first global convergence result for variational inference. We accomplish this in the context of an increasingly popular alternative objective for variational inference, the expected forward KL divergence:

$$L_P(\phi) := \mathbb{E}_{P(X)} \text{KL} \left[ P(\Theta \mid X) \,\|\, Q(\Theta; f(X; \phi)) \right]. \tag{1}$$

Here, $P(X)$ denotes a marginal of the model and $P(\Theta \mid X)$ denotes the posterior. For each $x \in \mathcal{X}$, the approximation $Q(\Theta; \eta)$ to $P(\Theta \mid X = x)$ is indexed by the distributional parameters $\eta \in \mathcal{Y} \subseteq \mathbb{R}^q$, which are themselves the output of a neural network $f(x; \phi)$ with weights $\phi \in \Phi$. Approximating a posterior distribution by minimizing $L_P$ has a long history (Section 2.1), and is sometimes known as neural posterior estimation (NPE) (Papamakarios and Murray, 2016) and the "sleep" objective of reweighted wake-sleep (Bornschein and Bengio, 2015; Le et al., 2019). Minimization of this objective is straightforward: computing unbiased gradients requires only sampling $\theta, x \sim P(\Theta, X)$ from the

38th Conference on Neural Information Processing Systems (NeurIPS 2024).

joint model (Section 2.1), which is readily accomplished by ancestral sampling. This approach is "likelihood-free" in that the density of $P(X \mid \Theta)$ need not be evaluated, and therefore expected forward KL minimization is more widely applicable than ELBO-based optimization, which requires a tractable likelihood function. Analysis of the amortized problem (i.e., optimizing an objective that averages over $P(X)$) is beneficial when considering the forward KL; for the non-amortized problem in which a single observation $x$ is considered, only biased estimates of the gradient of the forward KL can be obtained using self-normalized importance sampling, making convergence difficult to establish (Bornschein and Bengio, 2015; Le et al., 2019; Owen, 2013). Our analysis considers a functional form of variational objective $L_P$, given by

$$L_F(f) := \mathbb{E}_{P(X)} \mathrm{KL}\left[P(\Theta \mid X) \mid\mid Q(\Theta; f(X))\right], \tag{2}$$

where $L_F : \mathcal{H} \to \mathbb{R}$ is defined over a general reproducing kernel Hilbert space of functions $\mathcal{H}$. We refer to (1) as the "parametric objective", as its argument is the parameters $\phi \in \Phi$, and we refer to (2) as the "functional objective" as its argument is a function $f \in \mathcal{H}$. These objectives are closely related: under a given network parameterization, provided $f(\cdot; \phi) \in \mathcal{H}$, we have $L_P(\phi) = L_F(f(\cdot; \phi))$. The objective $L_P$ has been considered in several related works (see Section 2.1). The formulation of $L_F$ and the analysis of its minimizer relative to those of $L_P$ is our main contribution.

We first demonstrate strict convexity of the functional objective $L_F$ when $Q$ is parameterized as an exponential family distribution (Section 3). This implies the existence of a unique global optimizer $f^*$ of $L_F$ for a large class of variational families. Afterward, we analyze kernel gradient flow dynamics using the neural tangent kernel to show that minimization of $L_P$ results (asymptotically) in an empirical mapping $f$ that is at most $\epsilon$-suboptimal relative to $f^*$, provided a sufficiently flexible neural network is used to parameterize $f$ (Section 4). Together, these results imply that in the infinite-width limit, optimization of $L_P$ by gradient descent recovers a unique global solution.

Our analysis relies on fairly mild conditions, the most important of which are the positive definiteness of the neural tangent kernel and the structure of the variational family (e.g., an exponential family) (Section 6). Our proofs further assume a two-layer ReLU network architecture, but we conjecture that this assumption can be relaxed, and our experiments (Section 5) demonstrate global convergence for a wide variety of architectures. We illustrate that the minimization of $L_P$ converges to a global solution for problems with both synthetic and semi-synthetic data, and that finite network widths exhibit the behavior of the asymptotic regime (i.e., that of an infinitely wide network).

We further show that optimizing $L_P$ can produce better posterior approximations than likelihood-based ELBO methods, which suffer from convergence to shallow local optima. These results suggest, surprisingly, that a likelihood-free approach to inference can outperform likelihood-based approaches. Further, even for practitioners interested in inference for a single observation $x_0$, for whom amortization is not needed for computational efficiency, our approach may still be preferable to traditional ELBO-based inference due to the convergence guarantees of the former.

**Related work.** There is a large body of literature that analyzes the convergence of variational inference methods that target the ELBO. These works typically prove rates of convergence to the posterior (Zhang and Gao, 2020) or to a local optimum of the objective, as the ELBO is not amenable to global minimization because it is nonconvex (Domke, 2020; Domke et al., 2023; Kim et al., 2023). Liu et al. (2023) demonstrated nonconvexity of the ELBO in the context of small-object detection, and showed empirically that the expected forward KL was more robust to the pitfalls of numerical optimization. The nonconvexity of the variational objective has previously been addressed through workarounds such as convex relaxations (Fazelnia and Paisley, 2018) or iterated runs of the optimization routine to improve the quality of the local optimum (Altosaar et al., 2018). Our work differs from previous work in that our convergence result is global. Additionally, our approach is novel compared to related analyses because we consider the (arguably) more complicated problem of amortized inference, where the variational parameters are the weights of a neural network and are shared among observations.

## 2 Background

### 2.1 The Expected Forward KL Divergence

The expected forward KL objective is equivalent to the sleep-phase objective of Reweighted Wake-Sleep (RWS) (Bornschein and Bengio, 2015), and to the objective optimized by forward amortized variational inference (FAVI) (Ambrogioni et al., 2019). It is also a special case of the thermodynamic variational objective (TVO) (Masrani et al., 2019). Similar objectives have been referred to as neural posterior estimation (NPE) (Papamakarios and Murray, 2016; Papamakarios et al., 2019), though in these works the prior distribution, and thus the marginal $P(X)$, mutates during training.

Objectives based on the forward KL divergence generally result in variational posteriors that are overdispersed, a desirable property compared to reverse KL-based optimization (Le et al., 2019; Domke and Sheldon, 2018).

Unbiased gradient estimation for the parametric objective $L_P$ is straightforward. The outer expectation over $P(X)$ allows for gradients to be computed as

$$\nabla_\phi \mathbb{E}_{P(X)} \text{KL}\left[P(\Theta \mid X) \mid\mid Q(\Theta; f(X; \phi))\right] = -\mathbb{E}_{P(\Theta)} \mathbb{E}_{P(X \mid \Theta)} \nabla_\phi \log q(\Theta; f(X; \phi)),$$

where $q$ is the density of $Q$ with respect to Lebesgue measure (see Appendix B for details). Rewriting the left-hand side as an expectation over $P(\Theta)$ and $P(X \mid \Theta)$ by Bayes' rule illustrates that samples can be drawn from this model by ancestral sampling of $\Theta$ followed by $X$.

Other methods targeting the forward KL, such as the wake-phase of RWS, often optimize over a different expectation, typically $\mathbb{E}_{X \sim \mathcal{D}} \text{KL}\left[P(\Theta \mid X) \mid\mid Q(\Theta; f(X; \phi))\right]$. Here, the outer expectation is over an empirical dataset $\mathcal{D}$ rather than $P(X)$. In this case, approximation techniques such as importance sampling are required to estimate the gradient, as sampling from $P(\Theta \mid X = x)$ is intractable for any $x$. Relying on importance sampling results in *biased* gradient estimates, with which stochastic gradient descent (SGD) may not converge (Bornschein and Bengio, 2015; Le et al., 2019).

### 2.2 The Neural Tangent Kernel

A neural network architecture and the parameter space $\Phi$ of its weights together define a family of functions $\{f(\cdot; \phi) : \phi \in \Phi\}$. Let $\ell(x, f(x))$ denote a general real-valued loss function and consider selecting the parameters $\phi$ to minimize $\mathbb{E}_{P(X)} \ell(X, f(X; \phi))$, where $P(X)$ is a distribution on the data space $\mathcal{X}$. The neural tangent kernel (NTK) (Jacot et al., 2018) analyzes the evolution of the function $f(\cdot; \phi)$ while $\phi$ is fitted by gradient descent to minimize the above objective. Continuous-time dynamics are used in the formulation; $\phi(t)$ and $f(\cdot; \phi(t))$ are defined for continuous time $t$. The parameters $\phi$ thus follow the ODE

$$\dot{\phi}(t) = -\nabla_\phi \mathbb{E}_{P(X)} \ell(X, f(X; \phi(t))). \tag{3}$$

Here, $\dot{\phi}$ denotes the derivative with respect to $t$, and by the chain rule, the function values $f(x; \phi(t))$ evolve via

$$\dot{f}(x; \phi(t)) = -\mathbb{E}_{P(X)} \underbrace{J_\phi f(x; \phi(t)) J_\phi f(X; \phi(t))^\top}_{\text{NTK}} \ell'(X, f(X; \phi(t))).$$

We define $\ell'(X, f(X)) := \nabla_f \ell(X, f(X))$ to simplify the notation. The product of Jacobians above is known as the *neural tangent kernel* (NTK):

$$K_\phi(x, x') = J_\phi f(x; \phi) J_\phi f(x'; \phi)^\top. \tag{4}$$

The seminal work of Jacot et al. (2018) defined and studied this kernel and established the convergence of $K_\phi$ to a limiting kernel for certain neural network architectures as the layer width grows large.

### 2.3 Vector-Valued Reproducing Kernel Hilbert Spaces

Most existing NTK-based analyses consider neural networks with scalar outputs and squared error loss. Instead, we consider neural networks with multivariate outputs to accommodate the multidimensional

distributional parameter $\eta$, which parameterizes our variational distribution $Q(\Theta; \eta)$. Furthermore, we consider the objective functions $L_P$ and $L_F$. Consequently, we rely on results from the vector-valued reproducing kernel Hilbert space (RKHS) literature, as these spaces contain vector-valued functions such as the network function $f(\cdot; \phi)$. Carmeli et al. (2008, 2006) provide a detailed review, and in the following, we summarize the key properties.

Recall that $\eta = f(\cdot; \phi)$ and $\eta \in \mathbb{R}^q$. An $\mathbb{R}^q$-valued kernel on $\mathcal{X}$ is a map $K : \mathcal{X} \times \mathcal{X} \to \mathbb{R}^{q \times q}$. The neural tangent kernel (4) is precisely such a kernel. If the $\mathbb{R}^q$-valued kernel $K$ is positive definite, then $K$ defines a unique Hilbert space of functions $\mathcal{H}$ whose elements are maps from $\mathcal{X}$ to $\mathbb{R}^q$ (Carmeli et al., 2006). This kernel $K$ is called the reproducing kernel, and the corresponding space of functions is the RKHS associated with the kernel $K$.

In an $\mathbb{R}^q$-valued RKHS, the reproducing property takes on a more general form. For any $x \in \mathcal{X}$, $f \in \mathcal{H}$, and $v \in \mathbb{R}^q$,
$$f(x)^\top v = \langle f(x), v \rangle_{\mathbb{R}^q} = \langle f(\cdot), K(\cdot, x)v \rangle_{\mathcal{H}},$$
where $\langle \cdot, \cdot \rangle_{\mathcal{H}}$ is the inner product of the Hilbert space $\mathcal{H}$. For any fixed $x \in \mathcal{X}$ and $v \in \mathbb{R}^q$, $K(\cdot, x)v$ is a function mapping from $\mathcal{X} \mapsto \mathbb{R}^q$, as is required to be an element of $\mathcal{H}$.

## 3    Convexity of the Functional Objective

We now turn to the analysis of the functional objective $L_F$ given in Equation (2). We fix an RKHS $\mathcal{H}$ over which to minimize $L_F$ for now, specializing to the particular choice of $\mathcal{H}$ based on the neural tangent kernel subsequently. Let $\ell(x, f(x)) = \mathrm{KL}\left[P(\Theta \mid X = x) \mid\mid Q(\Theta; f(x))\right]$. The functional $L_F$ then has the form $L_F = \mathbb{E}_{P(X)}\ell(X, f(X))$; we will use this form subsequently for our neural tangent kernel analysis. Our first result shows that targeting $L_F$ is highly desirable theoretically: $L_F$ admits a unique global minimizer if the variational family $Q$ is an exponential family, as is common practice in VI.

**Lemma 1.** *Suppose that $Q(\Theta; \eta)$ is an exponential family distribution in minimal representation with natural parameters $\eta$, sufficient statistics $T(\theta)$, and density $q(\theta; \eta)$ with respect to Lebesgue measure $\lambda(\Theta)$. Then, for any observation $x \in \mathcal{X} \subseteq \mathbb{R}^d$, the loss function*

$$\ell(x, \eta) = \mathrm{KL}\left[P(\Theta \mid X = x) \mid\mid Q(\Theta; \eta)\right]$$

*is strictly convex in $\eta$, provided that $P(\Theta \mid X = x) \ll Q(\Theta; \eta) \ll \lambda(\Theta)$ for all $\eta \in \mathcal{Y} \subseteq \mathbb{R}^q$.*

A proof of Lemma 1, which follows quickly from the convexity of the log partition function in the natural parameter (Wainwright and Jordan, 2008, Proposition 3.1), is provided in Appendix A. Lemma 1 shows the strict convexity of the function $\ell$ in $\eta$. This implies the strict convexity of the functional $L_F(f) = \mathbb{E}_{P(X)}\ell(X, f(X))$ in $f$ by the linearity of expectation, which in turn implies the existence of at most one global minimizer.

**Corollary 1.** *Suppose that $Q(\Theta; \eta)$ is an exponential family distribution. Then, under the conditions of Lemma 1, the functional objective*

$$L_F(f) := \mathbb{E}_{P(X)}\mathrm{KL}\left[P(\Theta \mid X) \mid\mid Q(\Theta; f(X))\right]$$

*is strictly convex in $f$. Consequently, the set of global minimizers of $L_F$ is either a singleton set or empty.*

We will assume that the set of global minimizers is nonempty (so that the minimization of $L_F$ is well-posed) and let $f^*$ denote the global minimizer. We also assume that $||f^*||_{\mathcal{H}} < \infty$ so that $f^* \in \mathcal{H}$. Hereafter, we use the term "unique" to mean unique almost everywhere with respect to $P(X)$. Furthermore, in a slight abuse of notation, $f^*$ will denote the unique equivalence class of functions that minimize $L_F(f)$.

Whereas Lemma 1 establishes the convexity of the (non-amortized) forward KL divergence, Corollary 1 establishes the convexity of $L_F$, an amortized objective, in function space. The convexity of the functional objective $L_F$ holds regardless of the distribution chosen for the outer expectation by the same linearity argument. Choices other than $P(X)$, however, may not permit unbiased gradient estimation, as is the case for the wake-phase updates of RWS (Section 2.1).

# 4 Global Optima of the Parametric Objective

In practice, we must directly minimize $L_P$ rather than $L_F$, as optimizing the latter over the infinite-dimensional space $\mathcal{H}$ directly is not tractable. Thus, in the second phase of our analysis, we consider convergence to $f^*$ by minimizing the parametric objective $L_P$ with gradient descent. We define $\phi$ across continuous time as in Equation (3). Continuous-time dynamics simplify theoretical analysis; SGD with unbiased gradients follows a (noisy) Euler discretization of the continuous ODE (Santambrogio, 2017; Yang et al., 2021). Considering $X \sim P(X)$ for the outer expectation in both $L_P$ and $L_F$ is key in this context: this choice enables unbiased stochastic gradient estimation for $L_P$ (see Appendix B), whereas other choices require approximations that result in biased gradient estimates (see Section 2.1) and thus follow different gradient dynamics.

Analysis of the trajectories of the parametric objective $L_P$ throughout its minimization initially seems infeasible: the argument of this objective is the neural network parameters $\phi$, and even well-behaved loss functions such as the mean squared error (MSE) are nonconvex in these parameters. Nevertheless, neural tangent kernel (NTK)-based results enable analysis of $L_P$. We bridge the divide between the minimizers of the convex functional $L_F$ and the nonconvex objective $L_P$ using the limiting kernel, and show that in the large-width limit, the optimization path of $L_P$ converges arbitrarily close to $f^*$, the unique minimizer of the functional objective $L_F$.

**Theorem 1.** *Consider the width-$p$ scaled 2-layer ReLU network, evolving via the flow*

$$\dot{f}_t(x) = -\mathbb{E}_{P(X)} K^p_{\phi(t)}(x, X)\ell'(X, f_t(X)), \tag{5}$$

*where $f_t$ denotes $f(\cdot, \phi(t))$. Let $f^*$ denote the unique minimizer of $L = L_F$ from Lemma 1, and fix $\epsilon > 0$. Then, under conditions (C1)–(C4), (D1)–(D4), and (E1)–(E5), there exists $T > 0$ such that almost surely*

$$\left[\lim_{p \to \infty} L(f_T)\right] \leq L(f^*) + \epsilon. \tag{6}$$

Regularity conditions (C1)–(C4), (D1)–(D4), and (E1)–(E5) are provided in Appendices C, D, and E, respectively. We consider a scaled two-layer ReLU network architecture (further detailed in Appendix C) and use this simple architecture to prove the results as the network width $p$ tends to infinity. Our results may also be extended to multilayer perceptrons with other activation functions. Below, we briefly sketch the key ingredients needed to prove Theorem 1.

Recall the NTK $K^p_\phi$ from Equation (4), where we now let $p$ denote the network width. For certain neural network architectures, Jacot et al. (2018) show that as the network width $p$ tends to infinity, the neural tangent kernel becomes stable and tends (pointwise) towards a fixed, positive-definite limiting neural tangent kernel $K_\infty$.

Under suitable positivity conditions on the limiting kernel, we take the domain $\mathcal{H}$ of $L_F$ to be the RKHS with kernel $K_\infty$ (Section 2.3). Because $L_F$ has a unique minimizer $f^*$, under mild conditions on $K_\infty$, $f^*$ may be characterized as the solution obtained by following kernel gradient flow dynamics in $\mathcal{H}$, that is, the ODE given by

$$\dot{f}_t(x) = -\mathbb{E}_{P(X)} K_\infty(x, X)\ell'(X, f_t(X)).$$

In other words, starting from some function $f_0$, following the *limiting* NTK gradient flow dynamics above minimizes the functional objective $L_F$ for sufficiently large $T$.

**Lemma 2.** *Let $f^*$ denote the minimizer of $L_F$ from Lemma 1, and $\epsilon > 0$. Fix $f_0$, and let $K_\infty$ denote the limiting neural tangent kernel. Let $f_0$ evolve according to the dynamics*

$$\dot{f}_t(x) = -\mathbb{E}_{P(X)} K_\infty(x, X)\ell'(X, f_t(X)).$$

*Suppose that the conditions of Lemma 1 and (E1)-(E3) hold. Then, there exists $T > 0$ such that $L(f_T) \leq L(f^*) + \epsilon$, where $L$ is the loss functional of $L_F$.*

Appendix E enumerates regularity conditions (E1)–(E3) and provides a proof of Lemma 2. The characterization of $f^*$ in Lemma 2 clarifies how the analysis of the parametric objective $L_P$ will proceed. The gradient flow $L_P$ causes the network function to similarly evolve according to a kernel

gradient flow via the empirical neural tangent kernel, that is,

$$\dot{f}(x;\phi(t)) = -\mathbb{E}_{P(X)}K^p_{\phi(t)}(x,X)\ell'(X,f(X;\phi(t))),$$

as derived in Section 2.2. Comparison of the minimizers of $L_P$ and $L_F$ can be accomplished by comparing the two gradient flows above, i.e. kernel gradient flow dynamics that follow $K^p_{\phi(t)}$ and $K_\infty$, respectively. As $K^p_{\phi(t)} \to K_\infty$ (Appendix D), these trajectories should not differ greatly: for any fixed $T$, the functions obtained by following the kernel gradient dynamics with $K^p_{\phi(t)}$ and $K_\infty$ can be made arbitrarily close to one another, provided $p$ is sufficiently large. The proof of Theorem 1 first selects a $T$ using Lemma 2, and then bounds the difference in the trajectories on $[0, T]$ for sufficiently large width $p$ by convergence of the kernels $K^p_{\phi(t)} \to K_\infty$. Our proof differs from previous results in that it relies on *uniform* convergence of kernels (cf. Appendices C and D), enabling the analysis of population quantities such as $\mathbb{E}_{P(X)}\ell(X, f(X))$.

Theorem 1 proves convergence to an $\epsilon$-neighborhood of the global solution when optimizing $L_P$ despite the highly nonconvex nature of this optimization problem in the network parameters $\phi$. For sufficiently flexible network architectures, optimization of $L_P$ thus behaves similarly to that of $L_F$, which we have shown is a convex problem in the function space $\mathcal{H}$ in Section 3.

## 5 Experiments

Having established conditions under which global convergence is guaranteed, the main aim of our experiments is to demonstrate approximate global convergence in practice, even for scenarios where the conditions assumed in our proofs are not satisfied exactly. Section 5.1 demonstrates that finite-neuron layer widths used in practice approximate the limiting behavior well, while Section 5.2 and Section 5.3 utilize problem-specific network architectures for amortized inference. Our results suggest that there may exist weaker assumptions under which global convergence is still guaranteed.

### 5.1 Toy Example

We first assess whether the asymptotic regime of Theorem 1 is relevant to practice with layers of finite width. We use a diagnostic motivated by the lazy training framework of Chizat et al. (2019), which provides the intuition that in the limiting NTK regime, the function $f$ behaves much like its linearization around the initial weights $\phi_0$:

$$f(x;\phi) \approx f(x;\phi_0) + J_\phi f(x;\phi_0)(\phi - \phi_0). \tag{7}$$

Liu et al. (2020) prove that equality holds exactly in the equation above if and only if $f(x;\phi)$ has a constant tangent kernel (i.e., $K_\infty$). Therefore, similarity between $f$ and its linearization indicates that the asymptotic regime of the limiting NTK is approximately achieved. Note that even if $f$ is linear in $\phi$, as in the above expression, it may still be highly nonlinear in $x$.

We consider a toy example for which $||x||_2 = 1$. The generative model first draws a rotation angle $\Theta$ uniformly between 0 and $2\pi$, and then a rotation perturbation $Z \sim \mathcal{N}(0, \sigma^2)$, where we take $\sigma = 0.5$. Conditional on $\Theta$ and $Z$, the data $x$ is deterministic: $x = [\cos(\theta + z), \sin(\theta + z)]^\top$. This construction ensures that the data lie on the sphere $\mathbb{S}^1 \subset \mathbb{R}^2$, which guarantees the positivity of the limiting NTK for certain architectures (Jacot et al., 2018). We aim to infer $\Theta$ given a realization $x$, marginalizing over the nuisance latent variable $Z$. Our variational family $Q(\Theta; f(x))$ is a von Mises distribution, whose support is the interval $[0, 2\pi]$. This family is an exponential family distribution, allowing for the application of Lemma 1. The encoder network $f(\cdot; \phi)$ is given by a dense two-layer network (that is, one hidden layer) with rectified linear unit (ReLU) activation, which we study as the network width $p$ grows. The network outputs $f(x;\phi)$ parameterize the natural parameter $\eta$.

We fit the neural network $f(x;\phi)$ in two ways. First, we use SGD to fit the network parameters $\phi$ to minimize $L_P$. Second, we fit the linearization $f_{\text{lin}}(x;\phi) = f(x;\phi_0) + J_\phi f(x;\phi_0)(\phi - \phi_0)$ in $\phi$. We perform both of these fitting procedures for various widths $p$. For both settings, stochastic gradient estimation was performed by following the procedure in Appendix B. For evaluation, we fix $N = 1000$ independent realizations $x_1^*, \ldots, x_N^*$ from the generative model with underlying ground-truth latent parameter values $\theta_1^*, \ldots, \theta_N^*$, and evaluate the held-out negative log-likelihood (NLL), $-\frac{1}{N}\sum_{i=1}^N \log q(\theta_i^* \mid f(x_i^*; \phi))$, for both functions: $f(x;\phi)$ and $f_{\text{lin}}(x;\phi)$. Figure 1 shows

the evolution of the held-out NLL across the fitting procedure for three different network widths $p$: $64, 256$ and $1024$. The difference in quality between the linearizations and the true functions at convergence diminishes as the width $p$ grows; for $p = 1024$, the two are nearly identical, providing evidence that the asymptotic regime is achieved.

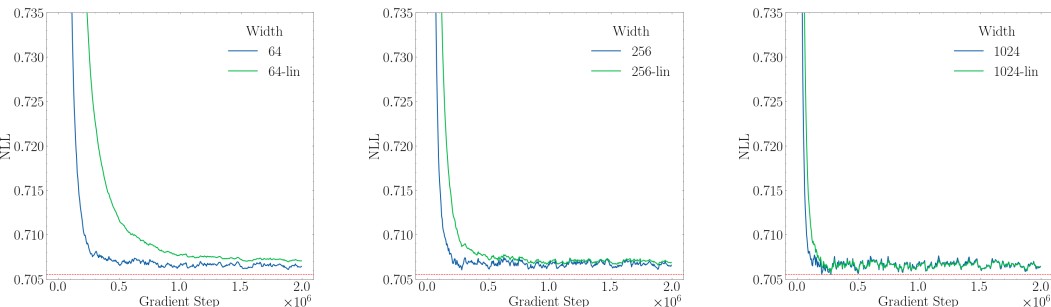

Figure 1: Negative log-likelihood across gradient steps, for network widths $64, 256$, and $1024$ neurons. NLL for the exact posterior is denoted by the red line.

## 5.2 Label Switching in Amortized Clustering

In this experiment and the subsequent one, we consider a more diverse set of network architectures. Although Theorem 1 assumes a shallow, dense network architecture, these experimental results show that empirically, global convergence can be obtained for many other architectures as well. We consider the difficult problem of amortizing clustering. In this problem, we are given an unordered collection (set) of data $x = \{x_1, \ldots, x_n\}$, $x_i \in \mathbb{R}$ and our objective is to infer the locations and labels of the cluster centers from which the data were generated. The generative model first draws a shift parameter $S \sim \mathcal{N}(0, 100^2)$ followed by

$$Z \mid (S = s) \sim \mathcal{N}\left([\mu_1 + s, \ldots, \mu_d + s]^\top, \sigma^2 I_d\right)$$

$$X_i \mid (Z = z) \overset{iid}{\sim} \sum_{j=1}^{d} p_j \mathcal{N}(z_j, \tau^2).$$

In the above, $\sigma^2, \tau^2, \mu \in \mathbb{R}^d$ and $p \in \mathbb{R}^d$ are known variances, locations, and proportions, respectively. The global parameter $S = s \in \mathbb{R}$ shifts the $d$ locations $\mu_1, \ldots, \mu_d$ to $\mu_1 + s, \ldots, \mu_d + s$. The cluster centers $Z$ are then obtained by adding noise to these locations. An implicit labeling is imposed on the centers by assigning each a dimension in $\mathbb{R}^d$ via the prior. Finally, the data are drawn independently from a mixture of $d$ univariate Gaussians with centers $Z_i$, $i = 1, \ldots, d$. We consider the tasks of inferring the scalar shift $S$ and the vector of cluster centers $Z$. We fix $\mu = [-20, -10, 0, 10, 20]^\top$ with $d = 5$. We fix the hyperparameters $\sigma = 0.5$ and $\tau = 0.1$, and artificially fix the shift as $S = 100$ to generate $n = 1000$ independent realizations $X = x$ from the generative model.

Inferring $S$ should be straightforward because the vector $\mu$ is known and fixed, ensuring that the joint likelihood $p(x, s)$ (marginalizing over $z$) is unimodal in $s$. However, inference on $Z$ may be difficult for likelihood-based methods because the order of the entries of $Z$ matters: the joint density $p(x, z, s) \neq p(x, \pi(z), s)$ for a permutation $\pi$, even though $p(x \mid z) = p(x \mid \pi(z))$. "Label switching" can thus pose a significant obstacle for likelihood-based methods, as any permutation of the cluster centers $Z$ will still explain the data well. This problem formulation thus results in a likelihood, and hence a posterior density, with many local optima but a single global optimum (i.e., where the cluster centers have the correct labeling).

Now we show that likelihood-based approaches to variational inference, such as maximizing the evidence lower-bound (ELBO), result in suboptimal solutions compared to the minimization of $L_P$. We take the variational distributions $q(S; f_1(x; \phi_1))$ and $q(Z; f_2(x; \phi_2))$ to be Gaussian. We fit the networks $f_1$ and $f_2$. Due to the exchangeability of the observations $x = \{x_1, \ldots, x_n\}$, we parameterize each as permutation-invariant neural networks, fitting both $\phi_1$ and $\phi_2$ to either minimize $L_P$ or to maximize the ELBO (see Appendix F). We perform 100 replications of this experiment across different random seeds, and consider two different parameterizations of the Gaussian variational

distribution: a mean-only parameterization with fixed unit variance and a natural parameterization with an unknown mean and unknown variance. Both of these variational families are exponential families, and so convexity (in the sense of Corollary 1) holds.

Figure 2 plots kernel-smoothed frequencies of point estimates of $S$, where each point estimate is the mode of the variational posterior for an experimental replicate. Both ELBO- and $L_P$-based training estimate $S = 100$ well. However, the limitations of ELBO-based training are evident in the fidelity of the posterior approximation of $Z$. Table 2 plots the average $\ell_1$ distance $||\hat{Z} - Z||_1$ between the variational mode $\hat{Z}$ and the true latent draw $Z$. Optimization of the ELBO converges to a local optimum that is a permutation of the entries of $Z$, resulting in a large $\ell_1$ distance on average. Minimization of $L_P$, on the other hand, converges to the global optimum without any label switching. Table 1 indicates the degree of label switching, showing the proportion of trials in which the entries $\hat{Z}$ were correctly ordered.

|         | ELBO | $L_P$ |
|---------|------|-------|
| Mean    | 0.03 | 1.00  |
| Natural | 0.02 | 1.00  |

Table 1: Proportion out of one hundred replicates where posterior mode of $q(Z; f_2(x; \phi_2))$ was a vector in increasing order.

|         | ELBO        | $L_P$     |
|---------|-------------|-----------|
| Mean    | 77.0 (45.2) | 1.8 (2.3) |
| Natural | 86.0 (26.9) | 2.9 (0.8) |

Table 2: Average $\ell_1$ distance $||\hat{Z} - Z||_1$ (and std. deviations) across one hundred replicates.

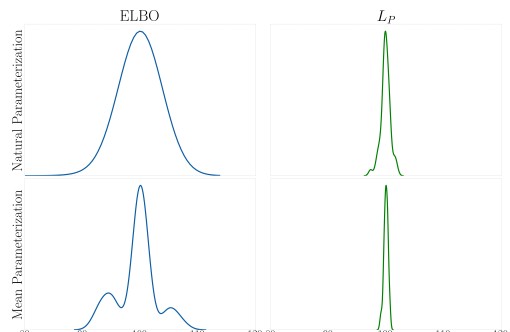

Figure 2: Mode of $q(S; f_1(x; \phi_1))$ across experimental replicates.

## 5.3 Rotated MNIST Digits

We consider the task of inferring a shared rotation angle $\Theta$ for a set of $N$ MNIST digits. The generative model is as follows. The rotation angle is drawn as $\Theta \sim \text{Uniform}(0, 2\pi)$, and for all $i \in [N]$ a noise term is drawn as $Z_i \sim \mathcal{N}(0_p, \sigma^2 I_p)$. Finally, each image is drawn as

$$X_i \mid (Z_i = z_i, \Theta = \theta) \sim \mathcal{N}\left(\texttt{RotateMNIST}(z_i, \theta), \tau^2\right), \quad i \in [N]. \tag{8}$$

Here, `RotateMNIST` is fitted ahead of time and fixed throughout this experiment. Given a latent representation $z$ and angle $\theta$, `RotateMNIST` returns a $28 \times 28$ MNIST digit image rotated counterclockwise by $\theta$ degrees. (See Appendix F for additional experimental details.) We aim to fit the variational posterior $q(\Theta; f(x_1, \ldots, x_N; \phi))$, implicitly marginalizing over the nuisance latent variables $Z_1, \ldots, Z_N$. The true data $\{x_i\}_{i=1}^N$ are generated from the above model, with $N = 1000$ digits and an underlying counterclockwise rotation angle of $\theta = 260$ degrees. We provide visualizations of some of these digits in Figure 3. The variational distribution $q(\Theta; f(x_1, \ldots, x_N; \phi))$ is taken to be a von Mises distribution with natural parameterization, as in Section 5.1, although for this example the architecture of the encoder network makes $f$ invariant to permutations of the inputs $\{x_i\}_{i=1}^N$, to reflect the exchangeability of the data. We fit $f$ to minimize $L_P$ using 100,000 iterations of SGD.

We compare to fitting $\theta$ to directly maximize the likelihood of the data $p(\theta, \{x_i\}_{i=1}^N)$. As this quantity is intractable, we maximize the importance-weighted bound (IWBO), which is a generalization of the ELBO (Burda et al., 2016), by maximizing the joint likelihood $p(\theta, \{z_i\}_{i=1}^N, \{x_i\}_{i=1}^N)$ in $\theta$ while fitting a Gaussian variational distribution on the variables $Z$.

The likelihood function is multimodal in $\theta$ due to the approximate rotational symmetry of several handwritten digits: zero, one, and eight are approximately invariant under rotations of 180 degrees. Additionally, the digits six and nine are similar following 180-degree rotations. These symmetries yield a multimodal posterior distribution on $\Theta$. Likelihood-based fitting procedures, such as maximizing the IWBO, often get stuck in these shallow local optima, while fitting $f$ to minimize the parametric objective $L_P$ finds a unique global solution. Figure 4 shows estimates of the angle $\theta$ conditional on the data $\{x_i\}_{i=1}^N$ during training with the IWBO objective, with $\theta$ initialized to a

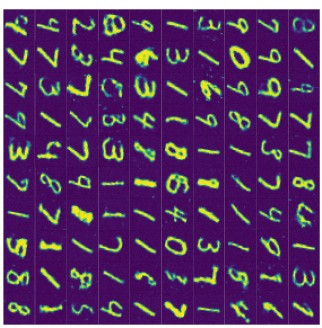

Figure 3: 100 of the $N = 1000$ data observations with counterclockwise rotation of 260 degrees.

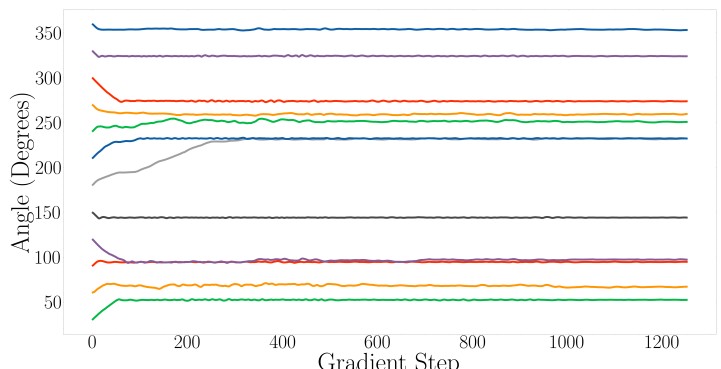

Figure 4: Estimate of angle $\theta$ across gradient steps, with fitting performed to maximize the IWBO.

variety of values. For some initializations, the IWBO optimization converges quickly to near the correct value of 260 degrees, but in many others, it converges to a shallow local optimum.

We perform the same routine, fitting $q(\Theta; f(x_1, \ldots, x_N; \phi))$ to minimize the expected forward KL divergence $L_P$. Figure 5 shows that across a variety of initializations of the angles, this approach always converges to a unique solution and fits the posterior mode to the correct value of 260 degrees. $L_P$ minimization converges rapidly, and so Figure 6 zooms in on the initial few thousand gradient steps to show the various trajectories among initializations.

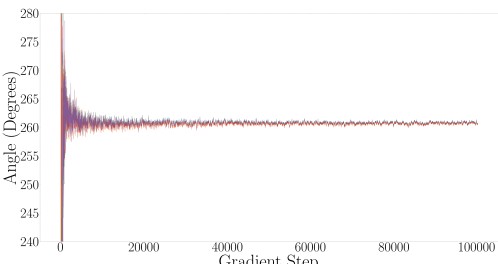

Figure 5: Mode of $q(\Theta; f(x_1, \ldots, x_N; \phi))$ across training (starting at different initializations) when minimizing objective $L_P$.

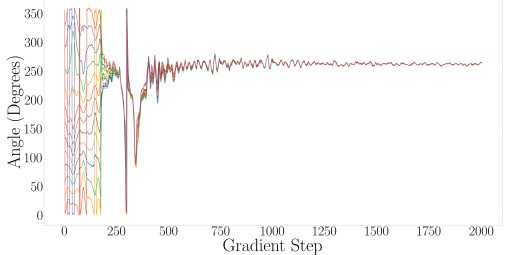

Figure 6: Zoomed-in trajectories across the first 2000 gradient steps, showing similar estimates regardless of initialization.

### 5.4 Local Optima vs. Global Optima

In this experiment, we directly compare the quality of the variational approximation minimizing the expected forward KL objective to that of the local optima found by optimizing the ELBO. We consider an adapted version of the rotated MNIST digit problem outlined above. Each digit $x_i$ is drawn from the model $x_i \sim \mathcal{N}\left(\texttt{Rotate}\left(x_i^0, \theta\right), \tau^2\right)$ for $i = 1, \ldots, 50$ with angle set to $\theta_{\text{true}} = 260$ degrees. The unrotated digits $x_i^0$ are fixed a priori, eliminating the nuisance latent variables $Z$ in this setting. This allows us to directly perform ELBO-based variational inference on $\Theta$, instead of merely maximizing the likelihood (or a bound thereof) as in Section 5.3. We fit an amortized Gaussian variational distribution with fixed variance $\sigma^2 = 0.5^2$ for inference on $\Theta$, and use the same prior as above. This is an exponential family with only one location parameter to be learned.

Fitting $q(\Theta; f(x_1, \ldots, x_{50}; \phi))$ is performed by minimizing either the negative ELBO or the expected forward KL divergence. The only differences between the two approaches are their objective functions: the data, network architecture, learning rates, and all other hyperparameters are the same. We optimize each objective function over 10,000 gradient steps, and measure the quality of the obtained variational approximations by a variety of metrics, including: the (non-expected) forward KL divergence; the reverse KL divergence; negative log-likelihood; and the angle point estimate. Intuition suggests that a global optimum should outperform local ones, and we find this to be the case. By any of

the performance metrics we consider, the global solution of the expected forward KL minimization outperforms the variational approximations found by optimizing the ELBO.

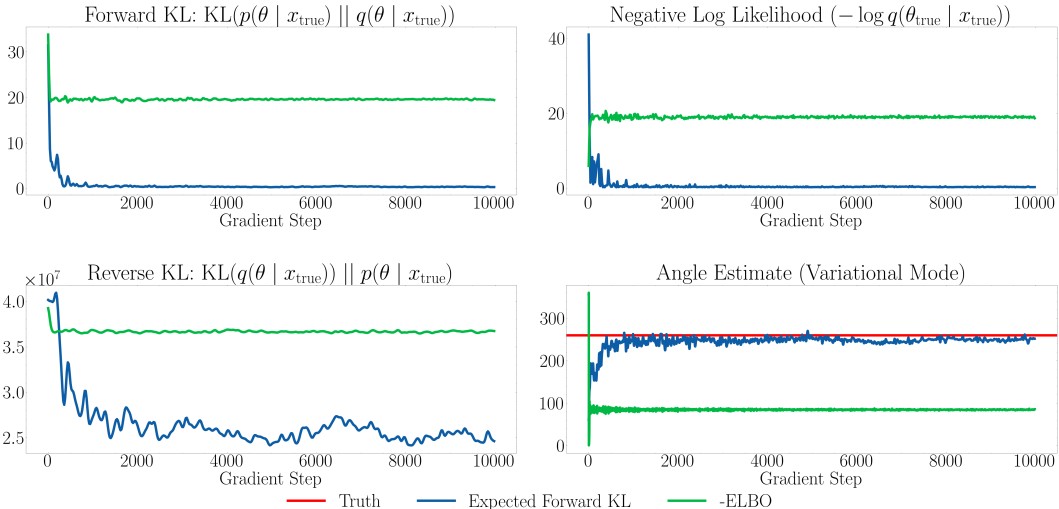

Figure 7: Forward and reverse KL divergences to the true posterior across fitting for minimization of the expected forward KL (blue) or the negative ELBO (green). We also plot the negative log likelihood of the true angle, as well as the variational mode (true angle $\theta_{\text{true}}$ is plotted in red.)

## 6 Discussion

In this work, we showed that in the asymptotic limit of an infinitely wide neural network, gradient descent dynamics on the expected forward KL objective $L_P$ converge to an $\epsilon$-neighborhood of a unique function $f^*$, a global minimizer. Our results depend on several regularity conditions, the most important of which is the positive definiteness of the limiting neural tangent kernel, the compactness of the data space $\mathcal{X}$, and the specific architecture considered, a single hidden-layer ReLU network. We conjecture and show experimentally that global convergence holds more generally. First, we illustrate that the asymptotic regime describes practice with a finite number of neurons (see Section 5.1). Second, our conditions allow for a wide variety of activation functions beyond ReLU (see Appendix D). Third, empirically, global convergence can be achieved with many network architectures. Beyond multilayer perceptrons, we illustrate global convergence for convolutional neural networks (CNNs) and permutation-invariant architectures in Section 5.2 and Section 5.3.

Expected forward KL minimization is a likelihood-free inference (LFI) method. For Bayesian inference, likelihood-based and likelihood-free methods are not typically viewed as competitors, but as different tools for different settings. In a setting where the likelihood is available, the prevailing wisdom suggests utilizing it. However, our results suggest that likelihood-free approaches to inference may be preferable even when the likelihood function is readily available. We find likelihood-based methods are prone to suffer the shortcomings of numerical optimization, often converging to shallow local optima of ELBO-based variational objectives. Expected forward KL minimization instead converges to a unique global optimum of its objective function.

There are situations in which ELBO optimization may nevertheless be preferable. First, if the likelihood function of the model is approximately convex and well-conditioned in the region of interest, ELBO optimization should recover a nearly global optimizer. Second, in certain situations, the ELBO can be optimized using deterministic optimization methods, which can be much faster than SGD. Third, if the generative model has free model parameters, with the ELBO they can be fitted while simultaneously fitting the variational approximation, with a single objective function for both tasks. Fourth, the ELBO can be applied with non-amortized variational distributions, which can have computational benefits in settings with few observations. Many important inference problems do not fall into any of these four categories. Even for those that do, the benefits of expected forward KL minimization, including global convergence, may outweigh the benefits of ELBO optimization.

## Acknowledgments

We thank the reviewers for their helpful comments and suggestions. This material is based on work supported by the National Science Foundation under Grant Nos. 2209720 (OAC) and 2241144 (DGE), and the U.S. Department of Energy, Office of Science, Office of High Energy Physics under Award Number DE-SC0023714.

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

## A  Convexity of the Functional Objective

We prove Lemma 1 from the manuscript below.

*Proof.* Let the log-density be given by $\log q(\theta; \eta) = \log h(\theta) + \eta^\top T(\theta) - A(\eta)$. First, observe that under the conditions given, the function $\ell$ is equivalent (up to additive constants) to a much simpler expression, the expected log-density of $q$, via

$$\ell(x, \eta) = \text{KL}\left[P(\Theta|X = x) \,\|\, Q(\Theta; \eta)\right]$$
$$= -\mathbb{E}_{P(\Theta|X=x)} \log q(\theta; \eta) + C,$$

where $C$ is a constant that does not depend on $\eta$. Now, the mapping $\eta \to -\log q(\theta; \eta)$ is convex in $\eta$ because its Hessian is $\frac{\partial A}{\partial \eta \partial \eta^\top} = \text{Var}(T(\theta)) \succ 0$ (cf. Chapter 6.6.3 of Srivastava et al. (2014)). Strict convexity follows from minimality of the representation (cf. Wainwright and Jordan (2008), Proposition 3.1). We can show $\ell$ is (strictly) convex in $\eta$ by applying linearity of expectation. We have for any $\lambda \in [0, 1]$

$$\ell(x, \lambda\eta_1 + (1-\lambda)\eta_2) = -\mathbb{E}_{P(\Theta|X=x)} \log q(\Theta; \lambda\eta_1 + (1-\lambda)\eta_2) \tag{9}$$
$$\leq -\mathbb{E}_{P(\Theta|X=x)}\left(\lambda \log q(\Theta; \eta_1) + (1-\lambda) \log q(\Theta; \eta_2)\right) \tag{10}$$
$$= \lambda\ell(x, \eta_1) + (1-\lambda)\ell(x, \eta_2) \tag{11}$$

where the second line follows from the convexity of the map $\eta \mapsto -\log q(\theta; \eta)$ above for any value of $\theta$. In fact, the inequality holds strictly as well by strict convexity above and the continuity of $\log q(\theta; \eta)$ in $\theta$. So the function $\ell(x, \eta)$ is strictly convex in $\eta$. $\square$

## B  Unbiased Stochastic Gradients for the Parametric Objective

Computation of unbiased estimates of the gradient of the loss function $L_P(\phi)$ with respect to the parameters $\phi$ is all that is needed to implement SGD for $L_P$. Under mild conditions (see Proposition 1), the loss function $L(\phi)$ may be equivalently written as

$$L_P(\phi) = \mathbb{E}_{P(X)}\mathbb{E}_{P(\Theta|X)} \log \frac{p(\Theta \mid X)}{q(\Theta; f(X; \phi))} = \mathbb{E}_{P(\Theta, X)} \log \frac{p(\Theta \mid X)}{q(\Theta; f(X; \phi))}$$

for density functions $p, q$, where $f(\cdot; \phi)$ denotes a function parameterized by $\phi$. Under the conditions of Proposition 1, differentiation and integration may be interchanged, so that

$$\nabla_\phi L_P(\phi) = \mathbb{E}_{P(\Theta, X)} \nabla_\phi \log \frac{p(\Theta \mid X)}{q(\Theta; f(X; \phi))} = -\mathbb{E}_{P(\Theta, X)} \nabla_\phi \log q(\Theta; f(X; \phi))$$

and unbiased estimates of the quantity can be easily attained by samples drawn $(\theta, x) \sim P(\Theta, X)$.

**Proposition 1.** *Let $(\Omega_1, \mathcal{B}_1)$, $(\Omega_2, \mathcal{B}_2)$ be measurable spaces on which the random variables $X : \Omega_1 \to \mathcal{X}$ and $\Theta : \Omega_2 \to \mathcal{O}$ are defined, respectively. Suppose that for all $x \in \mathcal{X}$ and all $\phi \in \Phi$ we have $P(\Theta \mid X = x) \ll Q(\Theta; f(x; \phi)) \ll \lambda(\Theta)$, with $\lambda(\Theta)$ denoting Lebesgue measure and $\ll$ denoting absolute continuity. Further, suppose that $\log\left(\frac{p(\Theta|X)}{q(\Theta; f(X;\phi))}\right)$ is measurable with respect to the product space $(\Omega_1 \times \Omega_2, \mathcal{B}_1 \times \mathcal{B}_2)$ for each $\phi \in \Phi$, and $\nabla_\phi \log q(\theta; f(x; \phi))$ exists for almost all $(\theta, x) \in \mathcal{O} \times \mathcal{X}$. Finally, assume there exists an integrable $Y$ dominating $\nabla_\phi \log q(\theta; f(x; \phi))$ for all $\phi \in \Phi$ and almost all $(\theta, x) \in \mathcal{O} \times \mathcal{X}$. Then for any $B \in \mathbb{N}$ and any $\phi \in \Phi$ the quantity*

$$\hat{\nabla}(\phi) = -\frac{1}{B}\sum_{i=1}^{B} \nabla_\phi \log q(\theta_i; f(x_i; \phi)), \quad (\theta_i, x_i) \overset{iid}{\sim} P(\Theta, X) \tag{12}$$

*is an unbiased estimator of the gradient of the objective $L_P$, evaluated at $\phi \in \Phi$.*

*Proof.* By the absolute continuity assumptions, for any $x \in \mathcal{X}$ the distributions $P(\Theta \mid X = x)$ and $Q(\Theta; f(x; \phi))$ admit densities with respect to Lebesgue measure denoted $p(\theta \mid x)$ and $q(\theta; f(x; \phi))$,

respectively. We may then rewrite the KL divergence from Equation (1) as

$$\text{KL}\left[P(\Theta \mid X = x) \mid\mid Q(\Theta; f(x;\phi))\right] := \mathbb{E}_{P(\Theta|X=x)} \log\left(\frac{dP(\Theta \mid X = x)}{dQ(\Theta; f(x;\phi))}\right)$$

$$= \mathbb{E}_{P(\Theta|X=x)} \log\left(\frac{p(\Theta \mid x)}{q(\Theta; f(x;\phi))}\right)$$

because the Radon-Nikodym derivative $dP/dQ$ is given by the ratio of these densities. Equation (1) is thus equivalent to

$$\mathbb{E}_{P(X)}\mathbb{E}_{P(\Theta|X)} \log\left(\frac{p(\Theta \mid X)}{q(\Theta; f(X;\phi))}\right) = \mathbb{E}_{P(\Theta,X)} \log\left(\frac{p(\Theta \mid X)}{q(\Theta; f(X;\phi))}\right).$$

This expectation is well-defined by the measurability assumption on $\log\left(\frac{p(\Theta|X)}{q(\Theta;f(X;\phi))}\right)$. To interchange differentiation and integration, it suffices by Leibniz's rule that the gradient of this quantity with respect to $\phi$ is dominated by a measurable r.v. $Y$. More precisely, there exists an integrable $Y(\theta, x)$ defined on the product space $\mathcal{O} \times \mathcal{X}$ such that $\left|\left|\nabla_\phi \log\left(\frac{p(\theta|x)}{q(\theta;f(x;\phi))}\right)\right|\right| \leq Y(\theta, x)$ for all $\phi \in \Phi$ and almost everywhere-$P(\Theta, X)$. This is assumed in the statement of the proposition, and so we have

$$\nabla_\phi \mathbb{E}_{P(\Theta,X)} \log\left(\frac{p(\Theta \mid X)}{q(\Theta; f(X;\phi))}\right) = \mathbb{E}_{P(\Theta,X)} \nabla_\phi \log\left(\frac{p(\Theta \mid X)}{q(\Theta; f(X;\phi))}\right)$$

$$= -\mathbb{E}_{P(\Theta,X)} \nabla_\phi \log q(\Theta; f(X;\phi))$$

and the result follows by sampling. $\square$

The variance of the gradient estimator can be reduced at the standard Monte Carlo rate, and for any $B$ Equation (12) can be used for SGD.

## C  The Limiting NTK

Before proceeding, we introduce the architecture specific to our analysis, a scaled two-layer network, and several theorems that we will use throughout the analysis.

The first result from Shapiro (2003) concerns optimization of the objective $f(x) = \mathbb{E}_{\xi \sim P} F(x, \xi)$ in $x$ via its empirical approximation $\hat{f}_n(x) = \frac{1}{n}\sum_{i=1}^n F(x, \xi_i), \xi_i \overset{iid}{\sim} P$. We reproduce this result below.

**Theorem 2** (Proposition 7 of Shapiro (2003)). *Let $C$ be a nonempty compact subset of $\mathbb{R}^n$ and suppose that (i) for almost every $\xi \in \Xi$ the function $F(\cdot, \xi)$ is continuous on $C$, (ii) $F(x, \xi)$, $x \in C$, is dominated by an integrable function, (iii) the sample $\xi_1, \dots, \xi_n$ is iid. Then the expected value function $f(x)$ is finite-valued and continuous on $C$, and $\hat{f}_n(x)$ converges to $f(x)$ with probability 1 uniformly on $C$.*

The next two results are integral forms of Gronwall's inequality that we use in subsequent analysis. We refer to Dragomir (2003) for a detailed review and summarize the results therein below.

**Theorem 3** (Gronwall's Inequality, Corollary 3 of Dragomir (2003)). *Let $u(t) \in \mathbb{R}$ be such that $u(t) \leq c_1 + c_2 \int_0^t u(s)ds$ for $t > 0$ and nonnegative $c_1, c_2$. Then*

$$u(t) \leq c_1 \exp[c_2 t].$$

**Theorem 4** (Theorem 57 of Dragomir (2003)). *Let $u(t) \in \mathbb{R}$ be such that $u(t) \leq c_1 + c_2 \int_0^t \int_0^s u(v)dvds$ for $t > 0$ and nonnegative $c_1, c_2$. Then*

$$u(t) \leq c_1 \exp[c_2 t^2/2].$$

Now we turn to specifics of the architecture we consider. Assume the function $f$ has the architecture of a (scaled) two-layer (single hidden layer) network mapping $f : \mathcal{X} \to \mathcal{Y}$ with $\mathcal{X} \subseteq \mathbb{R}^d$ and $\mathcal{Y} \subseteq \mathbb{R}^q$. We consider this network architecture for a given width $p$, and study each of the $i = 1, \dots, q$

coordinate functions of $f$. For a scaled two-layer network, the $i$th such function is

$$f(x;\phi)_i := \frac{1}{\sqrt{p}} \sum_{j=1}^{p} a_{ij}\sigma\left(x^\top w_j\right)$$

for $i = 1, \ldots, q$, where $\sigma$ denotes an activation function. The scaling depends on the width of the network $p$. The parameters $\phi$ are thus $\phi = \{w_j, a_{(\cdot),j}\}_{j=1}^{p}$ where $a_{(\cdot),j}$ denotes the vector $[a_{1j}, \ldots, a_{qj}]^\top$ (i.e. the $j$th coefficient for each component function $i$). The individual parameters have dimensions as follows: $w_j \in \mathbb{R}^d$ and $a_{(\cdot),j} \in \mathbb{R}^q$, for all $j = 1, \ldots, p$, where again $p$ denotes the network width and $d$ the data dimension dim $\mathcal{X}$. For ease hereafter, we write $a_j = a_{(\cdot),j}$ to refer to the entire $j$th vector of second layer network coefficients when the context is clear. As is standard, the first layer parameters are initialized as independent standard Gaussian random variables, i.e. $w_j \overset{iid}{\sim} \mathcal{N}(0, I_d)$ for all $j = 1, \ldots, p$. In other related works, the weight $a_{ij}$ is sometimes also drawn as $a_{ij} \overset{iid}{\sim} \mathcal{N}(0, 1)$ for all $i = 1, \ldots, q, j = 1, \ldots, p$, but in this work, we initialize these second-layer weights to zero for simplicity to ensure that at initialization, $f(\cdot; \phi) = 0$. A zero-initialized network function is used for analysis in several related works, e.g. Chizat et al. (2019) and Ba et al. (2020). For now, notationally we denote weights to be initialized as draws from an arbitrary distribution $D$, and we introduce specificity to $D$ as required.

The neural tangent kernel (Equation (4)) can be computed explicitly for this architecture and is given in the lemma below, which proves pointwise convergence to the limiting NTK at initialization as the width $p$ tends to infinity.

**Lemma 3** (Pointwise Convergence At Initialization). *For the architecture above, consider any $p$. Let $a \in \mathbb{R}^q, w \in \mathbb{R}^d$ be distributed according to $a, w \sim D$ for some distribution $D$ such that $a, w$ are integrable ($L_1$) random variables. Assume $\mathcal{X}$ is compact, and $\sigma'$ is bounded. Then, provided condition (C4) holds (see below), we have for any $x, \tilde{x} \in \mathcal{X}$ that*

$$K_{\phi(0)}^p(x, \tilde{x}) \overset{a.s.}{\to} \mathbb{E}_D K(x, \tilde{x}; w, a) \tag{13}$$

*as $p \to \infty$ where $K_{\phi(0)}^p$ denotes the NTK at initialization constructed from draws $a_j, w_j \overset{iid}{\sim} D$ and $K(x, \tilde{x}; w, a) \in \mathbb{R}^{q \times q}$ is the $q \times q$ matrix whose $k, l$th entry is given by*

$$\left[\mathbf{1}_{k=l}\sigma\left(x^\top w\right)\sigma\left(\tilde{x}^\top w\right) + a_k a_l \sigma'\left(x^\top w\right)\sigma'\left(\tilde{x}^\top w\right)\left(x^\top \tilde{x}\right)\right]$$

*for $k, l = 1, \ldots, q$.*

*Proof.* Consider the $k$th coordinate function of $f$. For any choice of $p$, the gradient is given by

$$\nabla_\phi f_k(x; \phi) = \begin{bmatrix} \frac{\partial f_k(x;\phi)}{\partial a_{k1}} \\ \vdots \\ \frac{\partial f_k(x;\phi)}{\partial a_{kp}} \\ \frac{\partial f_k(x;\phi)}{\partial w_1} \\ \vdots \\ \frac{\partial f_k(x;\phi)}{\partial w_p} \end{bmatrix} = \frac{1}{\sqrt{p}} \begin{bmatrix} \sigma\left(x^\top w_1\right) \\ \vdots \\ \sigma\left(x^\top w_p\right) \\ a_{k1}\sigma'\left(x^\top w_1\right)x \\ \vdots \\ a_{kp}\sigma'\left(x^\top w_p\right)x \end{bmatrix}$$

where we have imposed an arbitrary ordering on the parameters. In the above, we omitted partial derivatives $\frac{\partial f_k}{\partial a_{lj}}$ for $l \neq k, j = 1, \ldots, p$ because these are all identically zero. From this, it follows that for any fixed $x, \tilde{x} \in \mathcal{X}$, the $k, l$-th entry of $K_{\phi(0)}^p(x, \tilde{x})$ is given by

$$\nabla_\phi f_k(x; \phi(0))^\top \nabla_\phi f_l(\tilde{x}; \phi(0)) = \frac{1}{p}\sum_{j=1}^{p} \mathbf{1}_{k=l}\sigma\left(x^\top w_j\right)\sigma\left(\tilde{x}^\top w_j\right) +$$

$$\frac{1}{p}\sum_{j=1}^{p} a_{kj}a_{lj}\sigma'\left(x^\top w_j\right)\sigma'\left(\tilde{x}^\top w_j\right)\left(x^\top \tilde{x}\right).$$

The existence of the limiting NTK follows immediately: for each of the two terms above, each term is integrable by the compactness of $\mathcal{X}$ and domination (see (C4)). It follows that $K_\infty(x, \tilde{x})$ is the $q \times q$ matrix whose $k, l$th entry is given by

$$\mathbb{E}_{w,a\sim D}\left[\mathbf{1}_{k=l}\sigma\left(x^\top w\right)\sigma\left(\tilde{x}^\top w\right) + a_k a_l \sigma'\left(x^\top w\right)\sigma'\left(\tilde{x}^\top w\right)\left(x^\top \tilde{x}\right)\right]$$

with $w, a \sim D$. Convergence in probability pointwise follows from the weak law of large numbers, and almost sure convergence holds by the strong law of large numbers. $K(x, x; a, w)$ is integrable by the assumption (C4) (see below), so the expectation is well-defined. $\square$

The proof of the existence and pointwise convergence to the limiting NTK $K_\infty$ above is rather straightforward, and this result has been previously established in other works (Jacot et al., 2018). For our analysis of kernel gradient flows in Theorem 1 for the expected forward KL objectives $L_P$ and $L_F$, however, we require *uniform* convergence to $K_\infty$ over the entire data space $\mathcal{X}$.

We establish conditions under which this uniform convergence holds in two results, Proposition 2 and Proposition 3. Proposition 2, given below, concerns convergence at initialization to the limiting neural tangent kernel $K_\infty$ (i.e. before beginning gradient descent). Proposition 3, proven in Appendix D, demonstrates that across a finite training interval $[0, T]$, the NTK changes minimally from its initial value in a large width regime. Generally, we refer to the first result as "deterministic initialization" and the second as "lazy training" following related works (Jacot et al., 2018; Chizat et al., 2019).

Below, we give suitable regularity conditions and state and prove Proposition 2.

(C1) The data space is $\mathcal{X}$ is compact.

(C2) The distribution $D$ is such that $w \sim \mathcal{N}(0, I_d)$ and $a = 0$ w.p. 1. For $j = 1, \ldots, p$ iid draws from this distribution, we thus have $w_j \overset{iid}{\sim} \mathcal{N}(0, I_d)$ and $a_{ij} = 0$ w.p 1 for all $i, j$.

(C3) The activation function $\sigma$ is continuous. Under (C2), this implies that the function $K(\cdot, \cdot; a, w)$ from Lemma 3 with $a, w \sim D$ is almost surely continuous.

(C4) The function $K(x, \tilde{x}; a, w)$ is dominated by some integrable random variable $G$, i.e. for all $x, \tilde{x} \in \mathcal{X} \times \mathcal{X}$ we have $||K(x, \tilde{x}; a, w)||_F \leq G(a, w)$ almost surely for integrable $G(a, w)$.

**Proposition 2.** *Fix a scaled two-layer network architecture of width $p$, and let $\Phi$ denote the corresponding parameter space. Initialize $\phi(0)$ as independent, identically distributed random variables drawn from the distribution $D$ in (C2). Let $K^p_{\phi(0)} : \mathcal{X} \times \mathcal{X} \to \mathbb{R}^{q \times q}$ be the mapping defined by* $(x, \tilde{x}) \mapsto K_{\phi(0)}(x, \tilde{x}) = J_\phi f(x; \phi(0)) J_\phi f(\tilde{x}; \phi(0))^\top$. *Then, provided conditions (C1)–(C4) hold, we have as $p \to \infty$ that*

$$\sup_{x, \tilde{x} \in \mathcal{X}} ||K^p_{\phi(0)}(x, \tilde{x}) - K_\infty(x, \tilde{x})||_F \overset{a.s.}{\to} 0, \tag{DI}$$

*where $K_\infty(x, \tilde{x}) := \mathrm{plim}_{p \to \infty} K^p_{\phi(0)}(x, \tilde{x})$ is a fixed, continuous kernel.*

*Proof.* The proof follows by direct application of Proposition 7 of Shapiro (2003). Precisely, we satisfy i) almost-sure continuity of $K(\cdot, \cdot; a, w)$ by (C3), ii) domination by (C4), and iii) the draws comprising $K^p_{\phi(0)}$ are iid by assumption. By this proposition, then, we have uniform convergence of $K^p_{\phi(0)}$ to $K_\infty$ and obtain continuity of $K_\infty$ as well. $\square$

## D  Lazy Training

Below, we prove several results that will aid in proving the "lazy training" result of Proposition 3 (see below). Given the same architecture as above in Appendix C and a fixed width $p$ and time $T > 0$, we will begin by bounding $||w_j(T) - w_j(0)||$ and $||a_{kj}(T) - a_{kj}(0)||, ||a_{lj}(T) - a_{lj}(0)||$ for all $k, l = 1, \ldots, q$ and all $j = 1, \ldots, p$. As in Appendix C, there are several conditions that we impose and use in the following results. (D1)–(D2) are identical to (C1)–(C2), repeated for clarity.

(D1) The data space is $\mathcal{X}$ is compact.

(D2) The distribution $D$ is such that $w \sim \mathcal{N}(0, I_d)$ and $a = 0$ w.p. 1. For $j = 1, \ldots, p$ iid draws from this distribution, we thus have $w_j \overset{iid}{\sim} \mathcal{N}(0, I_d)$ and $a_{ij} = 0$ w.p 1 for all $i, j$.

(D3) The function $\ell(x, \eta) = \mathrm{KL}\left[P(\Theta \mid X = x) \| Q(\Theta; \eta)\right]$ is such that $\ell'(x; \eta)$ is bounded uniformly for all $x$ and for all $\eta \in \{f(x; \phi(t)) : t > 0\}$ by a constant $\tilde{M}$, uniformly over the width $p$. We recall that this notation is shorthand for $\nabla_\eta \ell(x, \eta)$.

(D4) The activation function $\sigma(\cdot)$ is non-polynomial and is Lipschitz with constant $C$. Note that the Lipschitz condition implies $\sigma$ has bounded first derivative i.e. $|\sigma'(r)| \leq C$ for all $r \in \mathbb{R}$.

With these conditions in hand, we now prove several lemmas for individual parameters.

**Lemma 4** (Lazy Training of $w$). *For the width $p$ scaled two-layer architecture above, assume that conditions (D1)–(D4) hold. Let $\phi$ evolve according to the gradient flow of the objective $L_P$, i.e.*

$$\dot{\phi}(t) = -\nabla_\phi L_P(\phi).$$

*Fix any $T > 0$. Then for all $j = 1, \ldots, p$,*

$$||w_j(T) - w_j(0)||_2 \leq ||w_j(0)||_2 \cdot D_{p,T} + E_{p,T} \tag{14}$$

*almost surely, where $D_{p,T}, E_{p,T}$ are constants depending on $p, T$, and $\lim_{p \to \infty} D_{p,T} = 0$ and $\lim_{p \to \infty} E_{p,T} = 0$.*

*Proof.* First note that for any fixed $j$, we have

$$J_{w_j} f(x; \phi) = \begin{bmatrix} \nabla_{w_j} f_1(x; \phi)^\top \\ \vdots \\ \nabla_{w_j} f_q(x; \phi)^\top \end{bmatrix} = \frac{1}{\sqrt{p}} \begin{bmatrix} a_{1j} \sigma'\left(x^\top w_j\right) x^\top \\ \vdots \\ a_{qj} \sigma'\left(x^\top w_j\right) x^\top \end{bmatrix} \in \mathbb{R}^{q \times d}$$

as required, where $a_{ij} \in \mathbb{R}$ for $i = 1, \ldots, q$ and $x \in \mathcal{X} \subseteq \mathbb{R}^d$ from (D1). We can bound the operator 2-norm of this matrix by observing that for any $y \in \mathbb{R}^d$ we have

$$||J_{w_j} f(x; \phi) y||_2^2 = \frac{1}{p} \cdot \left(\sum_{i=1}^q a_{ij}^2\right) \cdot \sigma'\left(x^\top w_j\right)^2 \left(x^\top y\right)^2$$

$$\leq \frac{C^2}{p} ||a_j||_2^2 \cdot ||x||_2^2 \cdot ||y||_2^2 \quad \text{by (D4) and Cauchy-Schwarz}$$

$$\implies ||J_{w_j} f(x; \phi)||_2 \leq \frac{C}{\sqrt{p}} ||a_j||_2$$

by observing $||x||_2^2$ is bounded by (D1) (and we absorb this term into the constant $C$). By similar computations, we also have

$$J_{a_j} f(x; \phi) = \begin{bmatrix} \nabla_{a_j} f_1(x; \phi)^\top \\ \vdots \\ \nabla_{a_j} f_q(x; \phi)^\top \end{bmatrix} = \frac{1}{\sqrt{p}} \mathrm{diag} \begin{bmatrix} \sigma\left(x^\top w_j\right) \\ \vdots \\ \sigma\left(x^\top w_j\right) \end{bmatrix} \in \mathbb{R}^{q \times q}.$$

Using condition (D4), it follows that

$$||J_{a_j} f(x; \phi)||_2 \leq \frac{|\sigma(x^\top w_j)|}{\sqrt{p}}$$

$$\leq \frac{|\sigma(0)| + C|x^\top w_j|}{\sqrt{p}}$$

$$\overset{def}{=} \frac{K + C|x^\top w_j|}{\sqrt{p}}$$

$$\leq \frac{K + C||w_j||_2}{\sqrt{p}}$$

by Cauchy-Schwarz and (D1),(D4), where throughout the following we let $K := |\sigma(0)|$ and again $||x||$ terms are absorbed into the constant $C$. Now we will use these computations to bound the

variation on $w_j$ across the interval $(0,T]$. Fix any $t \in (0,T]$. Then we have

$$||w_j(t) - w_j(0)||_2 \leq \int_0^t ||\dot{w}_j(s)||ds$$

$$\leq \int_0^t \mathbb{E}_{P(X)}||J_{w_j}f(X;\phi(s))\ell'(X,f(X;\phi(s)))||_2 ds$$

$$\leq \tilde{M}\int_0^t \mathbb{E}_{P(X)}||J_{w_j}f(X;\phi(s))||_2 ds \text{ by (D3)}$$

$$\leq \frac{C\tilde{M}}{\sqrt{p}}\int_0^t ||a_j(s)||_2 ds \text{ by above work}$$

$$\overset{a.s.}{=} \frac{C\tilde{M}}{\sqrt{p}}\int_0^t ||a_j(s) - a_j(0)||_2 ds \text{ by (D2)}$$

$$\leq \frac{C\tilde{M}}{\sqrt{p}}\int_0^t \int_0^s ||\dot{a}_j(v)||_2 dvds$$

$$\leq \frac{C\tilde{M}}{\sqrt{p}}\int_0^t \int_0^s \mathbb{E}_{P(X)}||J_{a_j}f(X;\phi)||_2||\ell'(X,f(X;\phi(v)))||_2 dvds$$

$$\leq \frac{C\tilde{M}^2}{\sqrt{p}}\int_0^t \int_0^s \mathbb{E}_{P(X)}||J_{a_j}f(X;\phi)||_2 dvds \text{ by (D3)}$$

$$\leq \frac{C\tilde{M}^2 K t^2}{2p} + \frac{C^2\tilde{M}^2}{p}\int_0^t \int_0^s ||w_j(v)||_2 dvds \text{ by above work}$$

$$\leq \frac{C\tilde{M}^2 K t^2}{2p} + \frac{C^2\tilde{M}^2}{p}\int_0^t \int_0^s ||w_j(v) - w_j(0)||_2 + ||w_j(0)||_2 dvds$$

$$= \frac{C\tilde{M}^2 K t^2}{2p} + \frac{C^2\tilde{M}^2 t^2}{2p}||w_j(0)||_2 + \frac{C^2\tilde{M}^2}{p}\int_0^t \int_0^s ||w_j(v) - w_j(0)||_2 dvds$$

$$\leq \frac{C\tilde{M}^2 K T^2}{2p} + \frac{C^2\tilde{M}^2 T^2}{2p}||w_j(0)||_2 + \frac{C^2\tilde{M}^2}{p}\int_0^t \int_0^s ||w_j(v) - w_j(0)||_2 dvds$$

$$= c_1 + \int_0^t \int_0^s c_2||w_j(v) - w_j(0)||_2 dvds$$

with $c_1 = \frac{C\tilde{M}^2 K T^2}{2p} + \frac{C^2\tilde{M}^2 T^2}{2p}||w_j(0)||_2$ and $c_2 = \frac{C^2\tilde{M}^2}{p}$. Note that even though $c_1$ depends on $T$, this is constant as $T$ is fixed. We write these quantities in this way to recognize a Gronwall-type inequality that we can use to bound the left hand side. Indeed, by direct application of Theorem 57 of Dragomir (2003) (see Theorem 4) we have that

$$||w_j(t) - w_j(0)||_2 \leq c_1 \exp\left[\int_0^t \int_0^s c_2 dvds\right]$$

$$= c_1 \exp\frac{c_2 t^2}{2}$$

$$= \left(\frac{C\tilde{M}^2 K T^2}{2p} + \frac{C^2\tilde{M}^2 T^2}{2p}||w_j(0)||_2\right)\exp\left[\frac{C^2\tilde{M}^2 t^2}{2p}\right].$$

giving the result for $t = T$ if we take $D_{p,T} = \frac{C^2\tilde{M}^2 T^2}{2p}\exp\left[\frac{C^2\tilde{M}^2 T^2}{2p}\right]$ and $E_{p,T} = \frac{C\tilde{M}^2 K T^2}{2p}\exp\left[\frac{C^2\tilde{M}^2 T^2}{2p}\right]$. Clearly these constants satisfy $\lim_{p\to\infty} D_{p,T} = 0$ and $\lim_{p\to\infty} E_{p,T} = 0$ for any fixed $T$.

$\square$

**Lemma 5** (Lazy Training of $a$). *Under the same conditions as Lemma 4, let $\phi$ evolve according to the gradient flow of problem $L_P$, i.e.*

$$\dot{\phi}(t) = -\nabla_\phi L_P(\phi).$$

*Fix any $T > 0$. Then we have for any $j$ that*

$$||a_j(T)||_2 \leq ||w_j(0)||_2 \cdot F_{p,T} + G_{p,T} \tag{15}$$

*almost surely, where $E_{p,T}$ and $F_{p,T}$ are constants depending on $p, T$ satisfying $\lim_{p \to \infty} E_{p,T} = 0$ and $\lim_{p \to \infty} F_{p,T} = 0$.*

*Proof.* We will use much of the same work as in Lemma 4. Namely, $||a_j(t)||_2 = ||a_j(t) - a_j(0)||_2$ almost surely by (D2), and thereafter for any $t \in (0, T]$ we have

$$||a_j(t) - a_j(0)||_2 \leq \int_0^t ||\dot{a}_j(v)||_2 ds$$

$$\leq \frac{1}{\sqrt{p}} \int_0^t \mathbb{E}_{P(X)} ||J_{a_j} f(X; \phi)||_2 ||\ell'(X, f(X; \phi(v)))||_2 ds$$

$$\leq \frac{\tilde{M}}{\sqrt{p}} \int_0^t \mathbb{E}_{P(X)} ||J_{a_j} f(X; \phi)||_2 ds$$

$$\leq \frac{K\tilde{M}t}{p} + \frac{\tilde{M}C}{p} \int_0^t ||w_j(s)||_2 ds \text{ by work in Lemma 4}$$

$$\leq \frac{K\tilde{M}t}{p} + \frac{\tilde{M}C}{p} \int_0^t ||w_j(s) - w_j(0)||_2 + ||w_j(0)||_2 ds$$

$$\leq \frac{K\tilde{M}t}{p} + \frac{\tilde{M}Ct}{p} ||w_j(0)||_2 + \frac{\tilde{M}C}{p} \int_0^t D_{p,s} ||w_j(0)||_2 + E_{p,s} ds \text{ by Lemma 4}$$

$$\leq \frac{K\tilde{M}t}{p} + \frac{\tilde{M}Ct}{p} ||w_j(0)||_2 + \frac{\tilde{M}C}{p} \int_0^t E_{p,s} ds + \frac{\tilde{M}C}{p} ||w_j(0)||_2 \int_0^t D_{p,s} ds$$

$$= ||w_j(0)||_2 \left( \frac{\tilde{M}Ct}{p} + \frac{\tilde{M}C}{p} \int_0^t D_{p,s} ds \right) + \left( \frac{K\tilde{M}t}{p} + \frac{\tilde{M}C}{p} \int_0^t E_{p,s} ds \right)$$

$$\overset{def}{=} ||w_j(0)||_2 \cdot F_{p,t} + G_{p,t}.$$

Clearly, these constants satisfy $\lim_{p \to \infty} F_{p,t} \to 0$ and $\lim_{p \to \infty} G_{p,t} \to 0$ (to see this, simply plug in the forms of $D_{p,s}$ and $E_{p,s}$ from Lemma 4 above) and we have the result by taking $t = T$.

$\square$

Now with these results in hand, we may state and prove Proposition 3, given below.

**Proposition 3.** *Under the same conditions as Proposition 2, fix any $T > 0$. For any $t \in (0, T]$ let $K^p_{\phi(t)} : \mathcal{X} \times \mathcal{X} \to \mathbb{R}^{q \times q}$ be the mapping defined by $(x, \tilde{x}) \mapsto K_{\phi(t)}(x, \tilde{x}) = J_\phi f(x; \phi(t)) J_\phi f(\tilde{x}; \phi(t))^\top$. Then provided conditions (D1)–(D4) hold, we have as $p \to \infty$ that*

$$\sup_{x, \tilde{x} \in \mathcal{X}, t \in (0, T]} ||K^p_{\phi(t)}(x, \tilde{x}) - K^p_{\phi(0)}(x, \tilde{x})||_F \overset{a.s.}{\to} 0. \tag{LT}$$

*Proof.* Let us examine the $k, l$th term of the $q \times q$ matrix given by $K^p_{\phi(t)}(x, \tilde{x}) - K^p_{\phi(0)}(x, \tilde{x})$ for fixed $x, \tilde{x}$, and some $t \in (0, T]$. The $k, l$th term is given by (see the work in Appendix C):

$$\frac{1}{p} \sum_{j=1}^{p} \mathbf{1}_{k=l} \left( \sigma \left( x^\top w_j(t) \right) \sigma \left( \tilde{x}^\top w_j(t) \right) \right) - \tag{16}$$

$$\left( \sigma \left( x^\top w_j(0) \right) \sigma \left( \tilde{x}^\top w_j(0) \right) \right) \tag{17}$$

$$+ \frac{1}{p} \sum_{j=1}^{p} \left( a_{kj}(t) a_{lj}(t) \sigma' \left( x^\top w_j(t) \right) \sigma' \left( \tilde{x}^\top w_j(t) \right) \left( x^\top \tilde{x} \right) \right) - \tag{18}$$

$$\left( a_{kj}(0) a_{lj}(0) \sigma' \left( x^\top w_j(0) \right) \sigma' \left( \tilde{x}^\top w_j(0) \right) \left( x^\top \tilde{x} \right) \right). \tag{19}$$

Above, we have explicitly made clear the dependence of the parameters on time, e.g. $w_j(t)$ vs. $w_j(0)$. We aim to show that the quantity above tends to zero as $p \to \infty$. We first prove this holds pointwise, and will consider the red and blue terms one at a time for a fixed $x, \tilde{x}$.

First consider the $j$th summand of the red term. We will bound its absolute value. If $k \neq l$, we're done, so assume $k = l$. We have for any $j$ that

$$\left| \sigma \left( x^\top w_j(t) \right) \sigma \left( \tilde{x}^\top w_j(t) \right) - \sigma \left( x^\top w_j(0) \right) \sigma \left( \tilde{x}^\top w_j(0) \right) \right|$$

$$= \left| \sigma \left( x^\top w_j(t) \right) \sigma \left( \tilde{x}^\top w_j(t) \right) - \sigma \left( x^\top w_j(t) \right) \sigma \left( \tilde{x}^\top w_j(0) \right) + \right.$$

$$\left. \sigma \left( x^\top w_j(t) \right) \sigma \left( \tilde{x}^\top w_j(0) \right) - \sigma \left( x^\top w_j(0) \right) \sigma \left( \tilde{x}^\top w_j(0) \right) \right|$$

$$\leq \left| \sigma \left( x^\top w_j(t) \right) \right| \cdot \left| \sigma \left( \tilde{x}^\top w_j(t) \right) - \sigma \left( \tilde{x}^\top w_j(0) \right) \right| + \left| \sigma \left( \tilde{x}^\top w_j(0) \right) \right| \cdot \left| \sigma \left( x^\top w_j(t) \right) - \sigma \left( x^\top w_j(0) \right) \right|$$

and by the Lipschitz assumption on $\sigma(\cdot)$ and Cauchy-Schwarz, we can bound the quantity above as follows

$$\leq (K + C||x||_2 ||w_j(t)||_2) \cdot C ||\tilde{x}||_2 ||w_j(t) - w_j(0)||_2 + (K + C||x||_2 ||w_j(0)||_2) \cdot C ||x||_2 ||w_j(t) - w_j(0)||_2$$

$$= C^2 ||w_j(t) - w_j(0)||_2 \left( 2\frac{K}{C} + ||w_j(t)||_2 + ||w_j(0)||_2 \right) \quad \text{since } ||x||_2, ||\tilde{x}||_2 \text{ are bounded by (D1)}$$

$$\leq C^2 ||w_j(t) - w_j(0)||_2 \left( 2\frac{K}{C} + ||w_j(t) - w_j(0)||_2 + 2||w_j(0)||_2 \right) \quad \text{by triangle inequality}$$

$$= 2CK ||w_j(t) - w_j(0)||_2 + C^2 ||w_j(t) - w_j(0)||_2^2 + 2C^2 ||w_j(0)||_2 ||w_j(t) - w_j(0)||_2$$

where again $||x||_2$ has been absorbed into the constant $C$ by (D1). Using Lemma 4, we can bound all terms above almost surely using $||w_j(0)||_2$ as follows.

$$\leq 2CK \left( D_{p,t} ||w_j(0)||_2 + E_{p,t} \right) + C^2 \left( D_{p,t} ||w_j(0)||_2 + E_{p,t} \right)^2 + 2C^2 \left( D_{p,t} ||w_j(0)||_2^2 + E_{p,t} ||w_j(0)||_2 \right)$$

$$= \left( 2C^2 D_{p,t} + C^2 D_{p,t}^2 \right) ||w_j(0)||_2^2 + \left( 2CK D_{p,t} + 2C^2 D_{p,t} E_{p,t} + 2C^2 E_{p,t} \right) ||w_j(0)||_2 + \left( 2CK E_{p,t} + C^2 E_{p,t}^2 \right)$$

Recalling that $w_j(0) \overset{iid}{\sim} \mathcal{N}(0, I_d)$, we have that $||w_j(0)||_2$ and $||w_j(0)||_2^2$ are integrable with expectations denoted $\mu$ and $\nu$, respectively. All our work has allowed us to show that

$$\left| \frac{1}{p} \sum_{j=1}^{p} \left( \sigma\left(x^\top w_j(t)\right) \sigma\left(\tilde{x}^\top w_j(t)\right) - \sigma\left(x^\top w_j(0)\right) \sigma\left(\tilde{x}^\top w_j(0)\right) \right) \right|$$

$$\leq \frac{1}{p} \sum_{j=1}^{p} \left( 2C^2 D_{p,t} + C^2 D_{p,t}^2 \right) ||w_j(0)||_2^2 + \left( 2CKD_{p,t} + 2C^2 D_{p,t} E_{p,t} + 2^2 CE_{p,t} \right) ||w_j(0)||_2$$

$$+ \left( 2CKE_{p,t} + C^2 E_{p,t}^2 \right)$$

$$\overset{a.s.}{\to} \left( \lim_{p \to \infty} 2C^2 D_{p,t} + C^2 D_{p,t}^2 \right) \nu + \left( \lim_{p \to \infty} 2CKD_{p,t} + 2C^2 D_{p,t} E_{p,t} + 2C^2 E_{p,t} \right) \mu$$

$$+ \left( \lim_{p \to \infty} 2CKE_{p,t} + C^2 E_{p,t}^2 \right)$$

$$= 0$$

by conditions on $D_{p,t}$ and $E_{p,t}$ from Lemma 4, the strong law of large numbers, and the classic result from analysis that $\lim_{n \to \infty} a_n b_n = (\lim_{n \to \infty} a_n)(\lim_{n \to \infty} b_n)$, provided both limits on the right hand side exist. Lastly, we can achieve the same result for the blue term quickly. Because $a_{ij}(0) = 0$ w.p. 1 by (D2), we have almost surely that

$$\frac{1}{p} \sum_{j=1}^{p} \left( a_{kj}(t) a_{lj}(t) \sigma'\left(x^\top w_j(t)\right) \sigma'\left(\tilde{x}^\top w_j(t)\right) \left(x^\top \tilde{x}\right) \right) - $$

$$\left( \cancel{a_{kj}(0) a_{lj}(0) \sigma'\left(x^\top w_j(0)\right) \sigma'\left(\tilde{x}^\top w_j(0)\right) \left(x^\top \tilde{x}\right)} \right)$$

$$\leq \frac{1}{p} \sum_{j=1}^{p} |a_{kj}(t)||a_{lj}(t)|| \sigma'\left(x^\top w_j(0)\right) ||\sigma'\left(\tilde{x}^\top w_j(0)\right) |||x||_2 ||\tilde{x}||_2$$

$$\leq \frac{C^2}{p} \sum_{j=1}^{p} |a_{kj}(t)||a_{lj}(t)|$$

$$\leq \frac{C^2}{p} \sum_{j=1}^{p} ||a_j(t)||_2^2$$

because for all $j$, we have $|a_{kj}|, |a_{kj}|$ are dominated by $||a_j||_2$. From here, we have by Lemma 5 that we can bound each term in the sum above by

$$\leq \frac{C^2}{p} \sum_{j=1}^{p} \left( ||w_j(0)||_2 F_{p,t} + G_{p,t} \right)^2$$

$$= \frac{C^2}{p} \sum_{j=1}^{p} F_{p,t}^2 ||w_j(0)||_2^2 + 2F_{p,t} G_{p,t} ||w_j(0)||_2 + G_{p,t}^2$$

$$\overset{a.s.}{\to} 0$$

as $p \to \infty$ by similar logic to the above. Together, these results combine to show that $|K_{\phi(t)}^p(x, \tilde{x})_{kl} - K_{\phi(0)}^p(x, \tilde{x})_{kl}| \overset{a.s.}{\to} 0$ as $p \to \infty$. As $k, l$ were arbitrary $k, l \in 1, \dots, q$, we have $||K_{\phi(t)}^p(x, \tilde{x}) - K_{\phi(0)}^p(x, \tilde{x})||_F \overset{a.s.}{\to} 0$. This establishes pointwise convergence for some fixed $t \in (0, T]$. Uniform convergence over all of $\mathcal{X} \times \mathcal{X}$ and all $t \in (0, T]$ follows easily in this case. Firstly, the numbers $D_{p,t}, E_{p,t}, F_{p,t}$, and $G_{p,t}$ are monotonic in $t$, so we can bound uniformly for all $t \in (0, T]$ by taking $t = T$ in the expressions above. Secondly, in our work above, our bounds on the red and blue terms were independent of the choice of point $(x, \tilde{x})$. More precisely, the supremum over $x, \tilde{x}$ can be accounted for in the bounds easily by observing that $\sup_{x, \tilde{x} \in \mathcal{X}, t \in (0, T]} ||K_{\phi(t)}^p(x, \tilde{x}) - K_{\phi(0)}^p(x, \tilde{x})||_F$

can be bounded above by

$$\leq \sup_{x,\tilde{x}\in\mathcal{X}} \left\| \frac{1}{p}\sum_{j=1}^{p} \mathbf{1}_{k=l}\left( \sigma\left(x^\top w_j(t)\right)\sigma\left(\tilde{x}^\top w_j(t)\right) \right) \right.$$

$$\left. - \left( \sigma\left(x^\top w_j(0)\right)\sigma\left(\tilde{x}^\top w_j(0)\right) \right) \right\|$$

$$+ \sup_{x,\tilde{x}\in\mathcal{X}} \left\| \frac{1}{p}\sum_{j=1}^{p} \left( a_{kj}(t)a_{lj}(t)\sigma'\left(x^\top w_j(t)\right)\sigma'\left(\tilde{x}^\top w_j(t)\right)\left(x^\top\tilde{x}\right) \right) \right.$$

$$\left. - \left( a_{kj}(0)a_{lj}(0)\sigma'\left(x^\top w_j(0)\right)\sigma'\left(\tilde{x}^\top w_j(0)\right)\left(x^\top\tilde{x}\right) \right) \right\|$$

$$\leq \sup_{x,\tilde{x}\in\mathcal{X}} \frac{1}{p}\sum_{j=1}^{p} \left(2C^2 D_{p,T} + C^2 D_{p,T}^2\right)\|w_j(0)\|_2^2 + \left(2CKD_{p,t} + 2C^2 D_{p,T}E_{p,T} + 2C^2 E_{p,T}\right)\|w_j(0)\|_2$$

$$+ \left(2CKE_{p,T} + C^2 E_{p,T}^2\right)$$

$$+ \sup_{x,\tilde{x}\in\mathcal{X}} \frac{C^2}{p}\sum_{j=1}^{p} F_{p,T}^2\|w_j(0)\|_2^2 + 2F_{p,T}G_{p,T}\|w_j(0)\|_2 + G_{p,T}^2$$

$$\overset{a.s.}{\to} 0$$

by the same work as above.

$\square$

# E   Kernel Gradient Flow Analysis

We rely on additional regularity conditions outlined below. We will consider the following three flows in our proof of Theorem 1 (for some choice of $p$):

$$\dot{f}_t(x) = -\mathbb{E}_{P(X)}K_{\phi(t)}^p(x,X)\ell'(X,f_t(X)) \tag{20}$$

$$\dot{g}_t(x) = -\mathbb{E}_{P(X)}K_\infty(x,X)\ell'(X,g_t(X)) \tag{21}$$

$$\dot{h}_t(x) = -\mathbb{E}_{P(X)}K_{\phi(0)}^p(x,X)\ell'(X,h_t(X)) \tag{22}$$

where $f_t$ is shorthand for $f(\cdot;\phi(t))$. The three flows above can be thought of as corresponding to $L_P$, $L_F$, and a "lazy" variant, respectively. The flow of $h_t$ is "lazy" because it follows the dynamics of a fixed kernel, the kernel at initialization. The flow of $g_t$ also follows a fixed kernel, but the limiting NTK $K_\infty$ instead. The flow of $f_t$ is that obtained in practice, where the kernel $K_{\phi(t)}^p$ changes continuously as the parameters $\phi(t)$ evolve in time. The flow in $h_t$ is used to bound the differences between $f_t$ and $g_t$ in the proof of Theorem 1. We now enumerate the regularity conditions.

- (E1) The functional $L_F(f)$ satisfies $\inf_f L_F(f) > -\infty$.
- (E2) The limiting NTK $K_\infty$ is positive definite (so that the RKHS $\mathcal{H}$ with kernel $K_\infty$ is well-defined).
- (E3) Under (E1) and (E2), the function $f^*$ minimizing $L_F$ satisfies $\|f^*\|_\mathcal{H} < \infty$, so that $f^* \in \mathcal{H}$.
- (E4) For any choice of $p$, we have for all $t, x$ that $\ell'(x;f_t(x)), \ell'(x;g_t(x))$, and $\ell'(x;h_t(x))$ are bounded by a constant $\tilde{M}$.
- (E5) The function $\ell$ is $\tilde{L}$-smooth in its second argument, i.e. $\|\ell'(x,\eta_1)-\ell'(x,\eta_2)\| \leq \tilde{L}\|\eta_1-\eta_2\|$.

We first prove Lemma 2 from the manuscript.

*Proof.* Let $f^* \in \operatorname{argmin} L_F(f)$, where $L_F(f)$ is the functional objective. Hereafter $L$ denotes $L_F$. Then $L(f^*) > -\infty$ by (E1). Then if $f_t$ evolves according to the kernel gradient flow (21) above (i.e.,

the flow with kernel $K_\infty$), we have (from the chain rule for Fréchet derivatives) that

$$\dot{L}(f_t) = \frac{\partial L}{\partial f_t} \circ \frac{\partial f_t}{\partial t}.$$

By definition, $\frac{\partial f_t}{\partial t} = \dot{f}_t$. We also have $\frac{\partial L}{\partial f_t} : h \mapsto \mathbb{E}_{P(X)} \ell'(X, f_t(X))^\top h(X)$. Plugging this in yields

$$\dot{L}(f_t) = \mathbb{E}_{X \sim P(X)} \ell'(X, f_t(X))^\top \left[ -\mathbb{E}_{X' \sim P(X)} K_\infty(X, X') \ell'(X', f_t(X')) \right]$$
$$= -\mathbb{E}_{X, X' \sim P} \ell'(X, f_t(X))^\top K_\infty(X, X') \ell'(X', f_t(X')) \leq 0$$

by the positiveness of the kernel $K_\infty$ (from (E2)). Now define $\Delta_t = \frac{1}{2} ||f_t - f^*||_{\mathcal{H}}^2$, where $\mathcal{H}$ is the vector-valued reproducing kernel Hilbert space corresponding to the kernel $K_\infty$ (see Carmeli et al. (2006) for a detailed review). We let $\langle \cdot, \cdot \rangle$ denote the inner product on the space $\mathcal{H}$. It follows that $\frac{\partial \Delta_t}{\partial f_t} : h \mapsto \langle f_t - f^*, h \rangle$. Then by the chain rule we have

$$-\dot{\Delta}_t = -\langle f_t - f^*, \dot{f}_t \rangle$$
$$= -\langle f_t - f^*, -\mathbb{E}_{P(X)} K_\infty(\cdot, X) \ell'(X, f_t(X)) \rangle$$
$$= \mathbb{E}_{P(X)} \ell'(X, f_t(X))^\top [f_t(X) - f^*(X)]$$
$$\geq \mathbb{E}_{P(X)} \ell(X, f_t(X)) - \ell(X, f^*(X))$$
$$= L(f_t) - L(f^*).$$

To go from the second to the third line, we used the reproducing property of the vector-valued kernel, the definition of inner product, and the linearity of integration. More precisely, the reproducing property (cf. Eq. (2.2) of Carmeli et al. (2006)) tells us for any functions $g, h$ and fixed $x$,

$$\langle g, K_\infty(\cdot, x) h(x) \rangle = g(x)^\top h(x)$$

and so the third line results from the second by exchanging the integral and inner product. In the second-to-last line, we used the convexity of $\ell$ in its second argument (from Lemma 1 of the manuscript). Now consider the Lyapunov functional given by

$$\mathcal{E}(t) = t \left[ L(f_t) - L(f^*) \right] + \Delta_t. \tag{23}$$

Differentiating, we have

$$\dot{\mathcal{E}}(t) = L(f_t) - L(f^*) + t\dot{L}(f_t) + \dot{\Delta}_t \leq 0$$

by the above work because i) $t\dot{L}(f_t) \leq 0$ and ii) $L(f_t) - L(f^*) + \dot{\Delta}_t \leq 0$, implying that $\mathcal{E}(t) \leq \mathcal{E}(0)$ for all $t$. Evaluating, thus

$$t \left[ L(f_t) - L(f^*) \right] + \Delta_t \leq \Delta_0$$
$$t \left[ L(f_t) - L(f^*) \right] \leq \Delta_0 - \Delta_t$$
$$t \left[ L(f_t) - L(f^*) \right] \leq \Delta_0 \quad \text{since } \Delta_t \geq 0$$
$$\left[ L(f_t) - L(f^*) \right] \leq \frac{1}{t} \Delta_0.$$

and so we have that there exists a sufficiently large $T$ such that $|L(f_T) - L(f^*)| \leq \epsilon$ as desired. $\quad\square$

Using this result and our previous results, we are now able to prove Theorem 1 from the manuscript.

*Proof.* We will examine the three gradient flows

$$\dot{f}_t(x) = -\mathbb{E}_{P(X)} K_{\phi(t)}^p(x, X) \ell'(X, f_t(X)) \tag{24}$$
$$\dot{g}_t(x) = -\mathbb{E}_{P(X)} K_\infty(x, X) \ell'(X, g_t(X)) \tag{25}$$
$$\dot{h}_t(x) = -\mathbb{E}_{P(X)} K_{\phi(0)}^p(x, X) \ell'(X, h_t(X)) \tag{26}$$

and establish the result by the triangle inequality, i.e.

$$|L(f_T) - L(f^*)| \leq |L(f_T) - L(g_T)| + |L(g_T) - L(f^*)|, \tag{27}$$

where $L$ denotes the functional objective $L_F$. Let $\epsilon > 0$. The flow in $h_t$ will be used to help bound the first term, but we begin with the second term. By Lemma 2, pick $T > 0$ sufficiently large such that $|L(g_T) - L(f^*)| \leq \epsilon/2$. Fix this $T$. This provides a suitable bound on the second term.

Turning to the first term, by continuity of $L(f)$ in $f$, there exists $\delta > 0$ such that $y \in B(g_T, \delta) \implies |L(y) - L(g_T)| \leq \epsilon/2$. We will show that there exists $P$ sufficiently large such that $p > P$ implies $||f_T - g_T|| \leq \delta$ almost surely, yielding the desired bound on the first term of the decomposition above. Throughout, $|| \cdot ||$ denotes the $L^2$ norm of a function with respect to probability measure $P$ (i.e. the marginal distribution of $P(X)$ from our joint model $P(\Theta, X)$).

To show that there exists sufficiently large $P$ such that $||f_T - g_T|| \leq \delta$, we use another application of the triangle inequality

$$||f_T - g_T|| \leq ||f_T - h_T|| + ||h_T - g_T||$$

and construct bounds on the two terms on the right hand side using Proposition 2 and Proposition 3. Observe first that by (C2)/(D2), at initialization we have almost surely that $f_0 = g_0 = h_0 = 0$. Also note that by continuity of $K_\infty$ (established in Lemma 3) on the compact domain $\mathcal{X} \times \mathcal{X}$ we have $\sup_{x,\tilde{x}} ||K_\infty(x, \tilde{x})||_2 < M$ for some $M$. Finally, note that by (E5) the function $\ell'(x, \eta)$ is Lipschitz in its second argument with constant $\tilde{L}$. Below, we let $|| \cdot ||_2$ denote the 2-norm of a vector or matrix, depending on the argument, and $|| \cdot ||_F$ the Frobenius norm of a matrix. For functions, as stated $|| \cdot ||$ denotes the $L^2$ norm with respect to measure $P(X)$, i.e. $||f||^2 = \int f(X)^\top f(X) dP(X)$. From here, we have

$$
\begin{aligned}
||g_T - h_T|| &\overset{a.s.}{=} ||(g_T - g_0) - (h_T - h_0)|| \\
&= \left|\left| \int_0^T \mathbb{E}_{P(X)} \left[ K_\infty(\cdot, X)\ell'(X, g_t(X)) - K_{\phi(0)}^p(\cdot, X)\ell'(X, h_t(X)) \right] dt \right|\right| \\
&\leq \int_0^T \mathbb{E}_{P(X)} ||K_\infty(\cdot, X)\ell'(X, g_t(X)) - K_{\phi(0)}^p(\cdot, X)\ell'(X, h_t(X))|| dt \\
&= \int_0^T \mathbb{E}_{P(X)} ||K_\infty(\cdot, X)\ell'(X, g_t(X)) - K_\infty(\cdot, X)\ell'(X, h_t(X)) + \\
&\qquad K_\infty(\cdot, X)\ell'(X, h_t(X)) - K_{\phi(0)}^p(\cdot, X)\ell'(X, h_t(X))|| dt \\
&\leq \int_0^T \mathbb{E}_{P(X)} ||K_\infty(\cdot, X) \left[ \ell'(X, g_t(X)) - \ell'(X, h_t(X)) \right] || + \\
&\qquad ||K_\infty(\cdot, X)\ell'(X, h_t(X)) - K_{\phi(0)}^p(\cdot, X)\ell'(X, h_t(X))|| dt \tag{28}
\end{aligned}
$$

Now, we note the following facts. Firstly, for any kernel $K$ that is uniformly bounded (i.e. $||K(x, y)||_2 \leq M$ for any $x, y$), the $L^2$ norm of the function $||K(\cdot, X)v||$ for fixed $X, v$ can be bounded by

$$||K(\cdot, X)v||^2 = \int v^\top K(Y, X)^\top K(Y, X)v \, dP(Y) \leq \int ||K(Y, X)||_2^2 ||v||_2^2 dP(Y) \leq M^2 ||v||_2^2$$

$$\implies ||K(\cdot, X)v|| \leq M ||v||_2$$

Secondly, we have again for any fixed $v$ and $X$ that

$$\left|\left| \left[ K_\infty(\cdot, X) - K^p_{\phi(0)}(\cdot, X) \right] v \right|\right|^2 = \int v^\top \left[ K_\infty(Y, X) - K^p_{\phi(0)}(Y, X) \right]^\top \left[ K_\infty(Y, X) - K^p_{\phi(0)}(Y, X) \right] v dP(Y)$$

$$\leq \int ||K_\infty(Y, X) - K^p_{\phi(0)}(Y, X)||_2^2 ||v||_2^2 dP(Y)$$

$$\leq \left( \sup_{x,y} ||K_\infty(y, x) - K^p_{\phi(0)}(y, x)||_F \right)^2 ||v||_2^2$$

$$\implies \left|\left| \left[ K_\infty(\cdot, X) - K^p_{\phi(0)}(\cdot, X) \right] v \right|\right| \leq \sup_{x,y} ||K_\infty(y, x) - K^p_{\phi(0)}(y, x)||_F \cdot ||v||_2$$

since the matrix (spectral) 2-norm is dominated by the Frobenius norm. Plugging these facts into Equation (28) above, we have

$$\leq \int_0^T \mathbb{E}_{P(X)} M \cdot ||\ell'(X, g_t(X)) - \ell'(X, h_t(X))||_2 + \sup_{x,y} ||K_\infty(x, y) - K^p_{\phi(0)}(x, y)||_F \cdot ||\ell'(X, h_t(X))||_2 dt$$

$$\leq \int_0^T \mathbb{E}_{P(X)} M\tilde{L} ||g_t(X) - h_t(X)||_2 + \tilde{M} \sup_{x,y} ||K_\infty(x, y) - K^p_{\phi(0)}(x, y)|| dt \text{ by (E4), (E5)}$$

$$\leq \tilde{M}T \sup_{x,y} ||K_\infty(x, y) - K^p_{\phi(0)}(x, y)||_F + M\tilde{L} \int_0^T \mathbb{E}_{P(X)} \sqrt{||g_t(X) - h_t(X)||_2^2} dt$$

$$\leq \tilde{M}T \sup_{x,y} ||K_\infty(x, y) - K^p_{\phi(0)}(x, y)||_F + M\tilde{L} \int_0^T \sqrt{\mathbb{E}_{P(X)} ||g_t(X) - h_t(X)||_2^2} dt \text{ by Jensen's inequality}$$

$$= \tilde{M}T \sup_{x,y} ||K_\infty(x, y) - K^p_{\phi(0)}(x, y)||_F + M\tilde{L} \int_0^T ||g_t - h_t|| dt$$

$$\implies ||g_T - h_T|| \leq \tilde{M}T \sup_{x,y} ||K_\infty(x, y) - K^p_{\phi(0)}(x, y)||_F \exp(M\tilde{L}T) \text{ by Gronwall's inequality (Theorem 3).}$$

By Proposition 2, there thus exists $P_1$ such that for all $p > P_1$ we have $||g_T - h_T|| \leq \frac{\delta}{2}$ almost surely. We proceed nearly identically for the term $||h_T - f_T||$. We need only note that for sufficiently large $p$, say $p > P_2$, we can bound $K^p_{\phi(0)}$ uniformly (almost surely) by a constant $A > M$. To see this, observe that by Proposition 2 we have that there exists almost surely a sufficiently large $P$ such that $\sup_{x,y} ||K_\infty(x, y) - K^p_{\phi(0)}(x, y)||_F < A - M$ and so by triangle inequality we have

$$\sup_{x,y} ||K^p_{\phi(0)}||_F \leq \sup_{x,y} ||K^p_{\phi(0)}(x, y) - K_\infty(x, y)||_F + ||K_\infty(x, y)||_F$$

$$\leq \sup_{x,y} ||K^p_{\phi(0)}(x, y) - K_\infty(x, y)||_F + \sup_{x,y} ||K_\infty(x, y)||_F$$

$$\leq A - M + M = A$$

Thereafter,

$$||h_T - f_T|| \overset{a.s.}{=} ||(h_T - h_0) - (f_T - f_0)||$$

$$= \left|\left| \int_0^T \mathbb{E}_{P(X)} \left[ K_{\phi(0)}^p(\cdot, X)\ell'(X, h_t(X)) - K_{\phi(t)}^p(\cdot, X)\ell'(X, f_t(X)) \right] dt \right|\right|$$

$$\leq \int_0^T \mathbb{E}_{P(X)}||K_{\phi(0)}^p(\cdot, X)\ell'(X, h_t(X)) - K_{\phi(t)}^p(\cdot, X)\ell'(X, f_t(X))||dt$$

$$\leq \int_0^T \mathbb{E}_{P(X)}||K_{\phi(0)}^p(\cdot, X)\ell'(X, h_t(X)) - K_{\phi(0)}^p(\cdot, X)\ell'(X, f_t(X))|| +$$

$$||K_{\phi(0)}^p(\cdot, X)\ell'(X, f_t(X)) - K_{\phi(t)}^p(\cdot, X)\ell'(X, f_t(X))||dt$$

$$\leq \int_0^T \mathbb{E}_{P(X)} A \cdot ||\ell'(X, h_t(X)) - \ell'(X, f_t(X))||_2 +$$

$$\sup_{x,y,t \in (0,T]} ||K_{\phi(0)}^p(x,y) - K_{\phi(t)}^p(x,y)||_F \cdot ||\ell'(X, f_t(X))||dt$$

$$\leq \tilde{M}T \sup_{x,y,t \in (0,T]} ||K_{\phi(0)}^p(x,y) - K_{\phi(t)}^p(x,y)||_F + A\tilde{L} \int_0^T \mathbb{E}_{P(X)}||h_t(X) - f_t(X)||_2 dt$$

and we can similarly switch from $\mathbb{E}_{P(X)}||h_t(X) - f_t(X)||_2$ to the $L^2$ norm $||h_t - f_t||$ as above using Jensen's inequality, yielding

$$\leq \tilde{M}T \sup_{x,y,t \in (0,T]} ||K_{\phi(0)}^p(x,y) - K_{\phi(t)}^p(x,y)||_F + A\tilde{L} \int_0^T ||h_t - f_t||dt$$

$$\implies ||h_T - f_T|| \leq \tilde{M}T \sup_{x,y,t \in (0,T]} ||K_{\phi(0)}^p(x,y) - K_{\phi(t)}^p(x,y)||_F \exp\left(A\tilde{L}T\right).$$

Clearly, by the same logic as the above there exists $P_3$ such that $p > P_3$ implies $\tilde{M}T \sup_{x,y,t \in (0,T]} ||K_{\phi(0)}^p(x,y) - K_{\phi(0)}^p(x,y)|| \exp(A\tilde{L}T) \leq \delta/2$ by Proposition 3. Then for all $p > \max(P_1, P_2, P_3)$, we have almost surely that $||h_T - f_T|| \leq \delta/2$. This completes the proof, as in this case we have by the triangle inequality that $||f_T - g_T|| \leq \delta$ and so $|L(f_T) - L(g_T)| \leq \epsilon/2$ by construction.

$\square$

# F    Experimental Details

Our code is publicly available at `https://github.com/declanmcnamara/gcvi_neurips`. We used PyTorch (Paszke et al., 2019) for our experiments in accordance with its license, and NVIDIA GeForce RTX 2080 Ti GPUs.

### F.1    Toy Example

Recall the generative model for this problem, given by the following:

$$\Theta \sim \text{Unif}[0, 2\pi]$$

$$Z \sim \mathcal{N}(0, \sigma^2)$$

$$X \mid (\Theta = \theta, Z = z) \sim \delta\left([\cos(\theta + z), \sin(\theta + z)]^\top\right).$$

The variable $\sigma$ is a hyperparameter of the model that we take to be $\sigma = 0.5$. The model is constructed in such a way that $x \in \mathbb{S}^1$ satisfies assumptions (C1) and (D1), respectively. One thousand pairs of data points $\{\theta_i, x_i\}_{i=1}^{1000}$ were generated independently from the model above and fixed as the dataset for which the ground-truth latent parameter values are known.

We constructed scaled, dense single hidden-layer ReLU networks of varying widths, with $2^j$ neurons for $j = 6, \ldots, 12$ with the same architecture as in Appendix C and the initialization described in condition (C2). All networks were trained to minimize the expected forward KL objective; stochastic gradients were estimated using batches of 16 independent simulated $(\theta, x)$ pairs from the generative model, and SGD was performed using the Adam optimizer with a learning rate of $\rho = 0.0001$. We employ a learning rate scheduler that scales the learning rate as $O(1/I)$, where $I$ denotes the number of iterations. All models were fitted for 200,000 stochastic gradient steps, and the execution time is less than one hour. The natural parameter for the von Mises distribution is parameterized as $\eta = f(x; \phi) + 0.0001$. This small perturbation must be added because $f(\cdot; \phi) = 0$ at initialization and because the value of $\eta = 0$ lies outside the natural parameter space for this variational family.

For the linearized neural network models, all training settings were the same except for the architecture. For these models, we first constructed neural networks as above for each width to compute the Jacobian evaluated at the initial weights $J_\phi(x; \phi_0)$. Thereafter, the model in $\phi$ is fixed as

$$f(x; \phi) = f(x; \phi_0) + J_\phi(x; \phi_0)(\phi - \phi_0)$$

where $\phi, \phi_0$ are flattened vectors of parameters from the neural network architectures. Using this linearized model above, the parameter $\phi$ is fitted by SGD as above.

The plots in Figure 1 of the manuscript are constructed by evaluating the average negative log-likelihood on the dataset at each iteration, i.e. for the fixed $n = 1000$ pairs of observations above, we evaluate the finite-sample loss for the expected forward KL divergence. Up to fixed constants, this quantity is given by

$$-\frac{1}{n} \sum_{i=1}^{n} \log q(\theta_i; f(x_i; \phi))$$

where $\phi$ is the current iterate of the parameters (either the neural network parameters or the flattened vector of parameters of the same size for the linearized model). The horizontal red line in Figure 1 is set at the value $-\frac{1}{n} \sum_{i=1}^{n} \log p(\theta_i \mid x_i)$, where $p$ denotes the exact posterior distribution (computed using numerical integration over a fine grid of evaluation points).

## F.2 Label Switching in Amortized Clustering

Recall the amortized clustering model from the manuscript, given by

$$S \sim \mathcal{N}(0, 100^2)$$

$$Z \mid (S = s) \sim \mathcal{N}\left( \begin{bmatrix} \mu_1 + s \\ \vdots \\ \mu_d + s \end{bmatrix}, \sigma^2 I_d \right)$$

$$X_i \mid (Z = z) \sim \sum_{j=1}^{d} p_j \mathcal{N}(z_j, \tau^2).$$

We take $\sigma = 1, \tau = 0.1$ and artificially fix $s = 100$ as our realization from the prior on $S$. Thereafter, we generate $N = 1000$ independent, identically distributed samples $\{x_i\}_{i=1}^{1000}$ from the model above, conditional on observing $S = 100$. The other non-random hyperparameters are the cluster centers $\mu = [-20, -10, 0, 10, 20]^\top$, as well as the mixture weights for the mixture of Gaussians, which are uniform, i.e. $p_j = \frac{1}{d}$ for all $j$. These draws constitute the "data" for this problem for which inference on $S$ is desired conditionally on the draws $x_1, \ldots, x_{1000}$; separately, we seek to infer the random variable $Z$ conditional on the data as well. Although $Z$ and $S$ are closely related, we impose a mean-field variational family $q(S, Z \mid X_1 = x_1, \ldots, X_N = x_n) = q(S \mid X_1 = x_1, \ldots, X_N = x_N)q(Z \mid X_1 = x_1, \ldots, X_N = x_N)$ for ease, with both components taken to be isotropic Gaussian distributions.

We use two different parameterizations for each of $q(S)$ and $q(Z)$. The mean parameterization fixes the variance at unity and aims to learn only the mean – in this case, the natural parameter of this family of distributions is simply the mean, so the network parameterizes the variational means $\mu_S, \mu_Z$ for each of these distributions. We also use a natural parameterization with an unknown mean and variance, and in this case, the natural parameterizations $\eta_S, \eta_Z$ are output by the networks,

which we denote $f(\cdot; \phi)$ and $f(\cdot; \psi)$ for $S$ and $Z$, respectively. Both amortized encoder networks $f(\cdot, \phi), f(\cdot; \psi)$ take the entire set of points $\{x_1, \ldots, x_{1000}\}$ as input. The architecture should thus be permutation-invariant, as the order of these points does not matter for inference on the latent quantities of interest. We accomplish this by using a set transformer architecture for the networks, which achieves permutation invariance using self-attention blocks (Lee et al., 2019).

Parameters are fitted to minimize $L_P$ using SGD with stochastic gradients estimated using simulated draws of the same size as the observed data. We use a learning rate of 0.001 for fitting both $q(Z)$ and $q(S)$, with schedules that decrease these as training progresses across 50,000 total gradient steps. We perform one hundred replicates of this experiment across different random seeds, each time generating a new dataset and refitting the networks. Accordingly, there is a high computational cost: using 10 parallel processes, the experiment takes about 8 hours to run. We compare $L_P$-based minimization to fitting both networks to maximize an evidence lower bound (ELBO), using the IWBO with $K = 10$ importance samples as the objective, but otherwise keeping all other hyperparameters the same. Figure 2 in the manuscript plots the mode of $q(S; f(x_1, \ldots, x_N; \phi))$ across one hundred replicates of the experiment with different random seeds. The estimate should be approximately 100 to match the ground truth, and both ELBO-based and $L_P$-based training perform well.

For inference on $Z$, we extract the mode $\hat{Z} \in \mathbb{R}^5$ as the point estimate and compute the $\ell_1$ distance to the true draw $Z = z$ for our dataset (which is known a priori because the data were generated synthetically). ELBO-based training illustrates label-switching behavior, converging to a vector which is a permutation of the entries of the true draw $z$, resulting in a large $\ell_1$ distance. $L_P$-based fitting does not suffer from this pathological behavior and instead converges rapidly to the global optimum.

### F.3 Rotated MNIST

We use the MNIST dataset, freely available from `torchvision` under the BSD-3 License[1] to fit a GAN generative model of MNIST digits. The simplistic GAN uses dense networks for both the discriminator and generator and is fit to the data with binary cross-entropy loss. Training was stopped when the generator produced realistic images, sufficient for our problem (see Figure 3).

The variational distribution $q(\Theta; f(x_1, \ldots, x_N; \phi))$ on the shared rotation angle is taken to be von Mises for minimization of $L_P$. We parameterize $q$ using its natural parameterization as in Appendix F.1. The encoder for $\Theta$ uses three Conv2D layers to increase the number of channels to 64 (with a kernel size of 3 and a stride length of 2), followed by a flattening layer and a three-layer dense network with LeakyReLU activations. Permutation invariance is imposed by a naive averaging step across all observations in the dataset $\{x_i\}_{i=1}^N$ that are provided to the network. The neural network architecture for this example thus differs significantly from the simple two-layer ReLU network, yet still demonstrates the same global convergence behavior.

We compare to a semi-amortized approach for maximizing the IWBO for this example. To compute the IWBO, the practitioner must posit a variational distribution on $z_i$ for each image $x_i$. This is because only the full joint likelihood $p(\theta, \{z_i\}_{i=1}^N, \{x_i\}_{i=1}^N)$ is readily available. Accordingly, for this method we construct a second network $f(\cdot; \psi)$ used to construct a variational distribution $q(Z_i; f(X_i; \psi))$ on $Z_i$ for a given $X_i$. This network is amortized over all images, yielding $q(Z_i; f(x_i; \psi))$ for any image $x_i$ in the dataset $\{x_i\}_{i=1}^N$. We parameterize these distributions on the latent representation as multivariate (isotropic) Gaussian with mean and log-scale parameters for each dimension the outputs of an encoder network. The architecture for this encoder consists of three Conv2D layers, each with kernel size 3 and stride of length 2. Across the three layers we increase the number of channels from a single channel (the input) up to 8 channels. After the convolutional layers, we perform a flattening layer followed a 2-layer dense network with ReLU activations. As the latent dimension for the data is known to be 64, the outputs of the encoder are 128-dimensional to parameterize the mean and log-scale across each dimension.

As there is only one rotation angle of interest we directly maximize the joint likelihood $p(\theta, \{z_i\}_{i=1}^N, \{x_i\}_{i=1}^N)$ in $\theta$ by targeting the IWBO. Conceptually, this approach is the same as placing a point mass variational family on $\Theta$, i.e., we simply initialize $\theta_0$ prior to training and update its value directly to maximize the IWBO. This approach thus fits the parameters $\{\psi, \theta_0\}$ of $q(Z_i; f(x_i; \psi))$ and $\delta_{\theta_0}$, the distributions on the latent representations and the angle, respectively.

---

[1] https://github.com/UiPath/torchvision/blob/master/LICENSE

These parameters are optimized using the Adam optimizer with initial learning rate 0.01 and square summable learning rate schedule. The angle parameter value is initialized at a variety of angles in different trials to produce Figure 4.

The advantageous marginalization properties of $L_P$-based fitting allow it to perform inference on $\theta$ *without performing inference on the latent representations* $Z_i$; thus, the distributions $q(Z_i; f(x_i; \psi))$ need not be fitted when using this approach. We use the Adam optimizer for all parameter updates with initial learning rate of 0.0001 (and a square summable learning rate schedule) for $L_P$-based training, which only fits the parameters $\phi$ of $q(\Theta; f(x_1, \ldots, x_N; \phi))$.

Minimization of $L_P$ trains for 100,000 gradient steps, although convergence is rapid, and so trajectories are truncated to produce Figure 4. Runs were performed in parallel across multiple processes for both methods and completed in less than one hour.

### F.4 Local Optima vs. Global Optima

For this experiment, the setting above is modified slightly to remove the nuisance latent variables $\{Z_i\}$ and put ELBO-based optimization on more equal footing with expected forward KL minimization. The data are generated as follows: for $i = 1, \ldots, 50$, we fix latents $z_i \overset{iid}{\sim} p(z)$, and $\theta_{\text{true}} = 260$ degrees is set artificially. The digits $x_i^0$ are computed via the generative adversarial network and thereafter fixed for the rest of the problem (the superscript denotes a rotation angle of zero degrees). With the unrotated digits fixed, the only random variables in the model are $\Theta$ and the rotated versions of the digits.

Conditioning on the observed data $\{x_i\}_{i=1}^{50}$, $\Theta$ is thus the only latent to be inferred. In this setting, the ELBO can compute the full likelihood function $p(\theta, \{x_i\}_{i=1}^{50})$ for any value of $\Theta = \theta$ (the same Uniform$(0, 2\pi)$ is placed on the angle). The simulation-based approach of expected forward KL minimization can proceed as usual: we draw $\Theta \sim$ Uniform$(0, 2\pi)$ and simulated digits are obtained by rotating the digits according to the drawn angle and adding noise.

The variational distribution, as stated in the text, is Gaussian with fixed variance instead of the von Mises distribution used in the local optima section. This is necessary for a one-to-one comparison between the two methods, as the von Mises distribution is not reparameterizable for ELBO-based training. The mean of the Gaussian is parameterized via the same neural network architecture above for fitting by both the ELBO and the expected forward KL. We use the Adam optimizer with learning rate 0.001 for 10,000 gradient steps.

For computing some of the metrics in Figure 7, we rely on approximations as these quantities are difficult to compute exactly. Letting $x_{\text{true}}$ denote the observed data $\{x_i\}_{i=1}^{50}$, the forward KL divergence KL$(p(\theta \mid x_{\text{true}}) \parallel q(\theta \mid x_{\text{true}}))$ is estimated using self-normalized importance sampling with $K = 1,000$ samples drawn from the prior as a proposal. The reverse KL divergence KL$(q(\theta \mid x_{\text{true}}) \parallel p(\theta \mid x_{\text{true}})$ is computed as the difference $\log p(x) - \text{ELBO}(q)$; the log-evidence is approximated via the importance weighted bound (IWBO) with $K = 1,000$.

