# OpenReview forum: "Globally Convergent Variational Inference"
_NeurIPS.cc/2024/Conference — NeurIPS 2024 poster_

### Official Review · Reviewer_gbq5 · 2024-06-25

**Soundness:** 3
**Presentation:** 2
**Contribution:** 4
**Rating:** 7
**Confidence:** 1

**Summary:**

The paper studies an alternative objective for variational inference, the expected forward KL divergence. Under some technical assumptions, convexity is shown, which facilitates global optimization. Moreover, a tractable surrogate objective is presented and it is shown that the approximation error can be made arbitrarily small. Finally, an experimental evaluation suggests that global convergence may even occur when the technical assumptions are violated.

**Strengths:**

The paper addresses a very interesting problem and the methodology is clever. The technical development seems very careful.

**Weaknesses:**

The paper is very technical and dense. It would be helpful to provide more intuition.

**Questions:**

1. How restrictive are the assumptions?
2. Can you explain the intuition more how your approach achieves convexity?
3. I understand you presented an example where the expected forward KL outperforms the standard ELBO. How does optimization of these two objectives compare more generally in practice? I imagine there are situations where optimizing the ELBO still yields better approximations to the actual posterior. Do you have any insights into these more practical aspects?

**Limitations:**

Technical limitations are stated but it would be helpful to discuss more prominently how restrictive the assumptions are.

---

> ### Author Rebuttal · Authors · 2024-08-06
>
> Thank you very much for your review. Based on your comments, we will add some clarifying sections to motivate the key intuition further. We aim to address your main points below.
>
> > *How restrictive are the assumptions?*
>
> We believe our assumptions are mild enough to apply to many practical settings, and are generally less restrictive than those found in related work. Several regularity conditions such as compactness of the data space $\mathcal{X}$, continuity of $\sigma$ and boundedness of $\sigma'$, and positivity of the limiting kernel $K_\infty$ are generally standard prerequisites for proving NTK-type results. Restricting the setting to a particular neural network architecture and initialization is also a standard approach to the problem.
>
> Our work stands out from previous works in the generality of the setting: we allow for network outputs of arbitrary dimension, and consider an objective function that is an expectation over an ``infinite dataset'', i.e. $L_F(f) = \mathbb{E}_{P(X)} \ell(X, f(X))$. Although we consider the forward KL integrand for $\ell$, our results actually apply to any $\ell$ that is convex in the network outputs $f(X)$. To our knowledge, existing NTK-based analyses all restrict themselves in some way, only considering i) scalar-valued outputs, ii) mean-squared error loss, or iii) and objective function over a finite training dataset.
>
> > *Can you explain the intuition more how your approach achieves convexity?*
>
> The main intuition for our work is that we can make use of convexity by conducting the analysis of the forward KL optimization problem in a function space. In parameter space, there is no hope of utilizing convexity arguments -- neural networks may have millions of parameters, and the objective functions used to fit them are highly non-convex in these parameters.
>
> When neural network functions are viewed simply as points in a more general Hilbert space of functions, though, the forward KL objective $L_F$ is in fact convex in its function argument -- this is the result of Corollary 1. The remainder of the analysis utilizes NTK-based analysis to show that the gradient dynamics in parameter space essentially mirror those in function space, i.e. optimization behaves as if we follow a convex objective to its global optimum.
>
> > *I imagine there are situations where optimizing the ELBO still yields better approximations to the actual posterior. Do you have any insights into these more practical aspects?*
>
> This is a good point, and we have updated the discussion section accordingly to discuss in more detail settings where using the ELBO may be more useful for practitioners. Choosing to optimize the ELBO can be advantageous in several situations: if the likelihood function of the model is (approximately) convex in the main region of interest, optimization should behave well, and in certain situations the ELBO can even be updated with non-stochastic gradients. In the case where the generative model is unknown and needs to be learned simultaneously with the variational approximation, the ELBO can be used as an objective to fit both the model and the variational distribution, which is appealing in practice. Finally, the ELBO can be optimized in a non-amortized fashion -- this may be preferable for simplicity, but this approach is still known to struggle with the same issue of converging to local optima.
>
> While these particular situations do arise, minimization of the forward KL can be applied in essentially any setting, and maintains its convexity guarantees even when the model is arbitrarily complex -- this approach can even be used in some settings where ELBO-based training is impossible. One large class of methods for which the ELBO is unusable for likelihood-free inference, where the generative model typically consists of a highly complex simulator that does not admit a tractable likelihood function.

---

> ### Comment · Reviewer_gbq5 · 2024-08-13
>
> I thank the authors for their response. I do not have further questions.

---

### Official Review · Reviewer_grF1 · 2024-07-04

**Soundness:** 3
**Presentation:** 3
**Contribution:** 3
**Rating:** 6
**Confidence:** 2

**Summary:**

This work addresses a common problem of non-convexity while approximating posterior distributions using variational inference. Although using ELBO as a variational objective is popular, this paper considers a forward KL(FKL) divergence objective. Its first main contribution is to show that when the variational family belongs to the exponential family of distributions, the FKL objective is strictly convex in the variational parameters. In particular, the paper parameterizes the exponential variational family by a neural network. Their second contribution is to show that under certain conditions on the neural architecture, the solution of the FKL objective is only \e-suboptimal to the global functional solution of the  FKL objective. The paper also demonstrates the efficacy of the proposed with the help of some experiments.

**Strengths:**

1. Using expected forward KL with the exponential family is simple and interesting when dealing with ELBO's non-convexity.
2. The idea of using an exception with P(X) also seems interesting, as it simplifies the efforts of sampling from the posterior while using the FKL objective. Is this idea novel? If yes, the authors should signpost it.
3. The connection between NTK and the gradient dynamics of the functional objective is also interesting and novel.
4. The paper is well written.

**Weaknesses:**

1. I think the authors underestimate their findings in Lemma 1, i.e., the convexity of the variational objective when forward KL is used with the exponential family. Can they demonstrate its efficacy on a toy problem that doesn’t use any neural network?
2. It is slightly misleading to say that the FKL objective with NTK finds global optima. Since the NN is highly non-convex, the result only shows that the local solution is close to the global one and that too in the limit of NN width.
3. I don't understand how Lemma 1 shows that L_F is strictly convex. I believe it just proves convexity.

**Questions:**

Minor comments.
L45: L_p to L_P
L771: What does \eta over equality mean?
L843: a,\in - remove

---

> ### Author Rebuttal · Authors · 2024-08-06
>
> Thank you for your review. Below, we attempt to address your main points.
>
> > *The idea of using an expectation with P(X) also seems interesting...is this idea novel?*
>
> Although it's not novel (e.g. Section 2.1), we do think it is underappreciated. Our main contribution lies in the analysis of this objective function. Previous motivations for using the expected forward KL divergence come from either 1) a setting where likelihood function is not available or 2) the practitioner's desire for conservative uncertainty quantification, a property of minimizers of this objective.
>
> Our analysis suggests a much stronger motivation for using the expected forward KL objective: it behaves like a convex objective and thus yields unique solutions, regardless of random seeds, initializations, etc. The shortcomings of numerical optimization are a major trouble point for use of VI over related methods such as MCMC in practice, and our work resolves many of these concerns. Beyond popularizing this choice of objective function as an alternative to the ELBO within the VI community, we hope that our results convince other Bayesians (e.g. practitioners of MCMC) of the validity of VI. We help resolve a major concern that one may end up with a local solution of unknown suboptimality.
>
> > *Can [the authors] demonstrate [Lemma 1]'s efficacy on a toy problem that doesn't use any neural network?*
>
> Lemma 1 in itself is interesting, but not practical. It illustrates the atypical paradigm of having a function that is well behaved (convex), but **not computable**. Neither $\ell(x, \eta)$ nor $\nabla_\eta \ell(x, \eta)$ are computable, or even unbiasedly estimable -- except in certain rare cases where the posterior itself is tractable, which of course makes VI unnecessary in the first place.
>
> On a non-amortized problem where one has a single $x$ and wishes to minimize $\ell(x, \eta)$ in $\eta$, the convexity result becomes an afterthought -- optimization of this objective cannot proceed because we cannot run SGD (we do not have access to unbiased gradients).
>
> The corollary implying that the amortized problem $L_F$ is convex is the foundation for the rest of our work. It implies the existence of a global minimum, and the remainder of our analysis shows the gradient dynamics of $L_P$ can converge arbitrarily close to this global optimum. A distinguishing feature of $L_P(\phi)$ is that we can estimate its gradients unbiasedly (e.g., as described in Appendix B).
>
> > *It is slightly misleading to say that the FKL objective with NTK finds global optima...the result only shows that the local solution is close to the global one.*
>
> Although asymptotic results such as ours will indeed only hold approximately in practice, we contend that many meaningful theoretical results take this form. Even some that might be regarded as standard results, such as the convergence of SGD with diminishing step size, are asymptotic results that are only approximated in practice with finite time and compute. Within the NTK literature in particular, asymptotic results of the nature of ours (i.e., in the limit of an arbitrarily wide network) are difficult to avoid due to the intractability of finite-width analysis. Although our results are asymptotic, we have made them as strong as possible (e.g., almost sure convergence), and have shown in our experiments that the asymptotic behavior (i.e., optimization trajectories similar to those in convex optimization) can actually be obtained in practice.
>
> You are correct that one will ultimately obtain a local solution in the parameter space. We show, though, that the "local solution" can be made arbitrarily close to the global minimizer $f^*$ by increasing the network width $p$ -- this allows one to obtain the ``de facto'' global minimizer. The ability to alter the width provides the user with a large amount of control over behavior of the optimization problem. Without our analysis, understanding the results of the optimization problem seems intractable: as you correctly point out, in parameter space the objective function is highly nonconvex, and *the degree of suboptimality* of the local solution is totally unknown for any given run of the optimization routine.
>
> Our result, on the other hand, bounds the degree of suboptimality, and more importantly shows that the user can shrink this bound by increasing the network width. We can always get $\epsilon$-close to the global optimum for any $\epsilon$. We show this result extends to practice for even modest network widths, and $\epsilon$ appears to be small enough on real problems that one i) converges to the same minimizer regardless of initialization and ii) this minimizer is practically indistinguishable from the true global minimum $f^*$ in the RKHS.
>
> >  *I don't understand how Lemma 1 shows that $L_F$ is strictly convex. I believe it just proves convexity.*
>
> Thank you for catching this; we have added a mild condition to Corollary 1 under which strict convexity holds. Previously, any two functions $f,g$ such that $f \neq g$ but $f = g$ almost surely would violate strict convexity because $L_F(f) = L_F(g)$ even though $f \neq g$. We have added the condition that the domain of $L_F$ is a RKHS $\mathcal{H}$ with respect to the measure $P(X)$. Under this assumption, two functions $f,g$ equal to each other almost surely on $P(X)$ are in fact regarded as the same element in the RKHS (they correspond to the same equivalence class of functions). With this condition, strict convexity holds as any $f \neq g$ in the RKHS differ on a set of nonzero measure, allowing the strict convexity inequality to extend to the integral.
>
> > *L771: What does $\eta$ over equality mean?*
>
> This notation meant "equality up to constants that do not depend on $\eta$", i.e. the right-hand side has the same gradient with respect to $\eta$ as the left-hand side. We have updated the notation to make this more clear and now instead use a generic constant $c$ on the right-hand side.

---

> > ### Comment · Reviewer_grF1 · 2024-08-12
> >
> > I thank the authors for satisfactorily responding to my queries.

---

### Official Review · Reviewer_KvcV · 2024-07-10

**Soundness:** 3
**Presentation:** 3
**Contribution:** 2
**Rating:** 5
**Confidence:** 4

**Summary:**

The authors established the global convergence of a particular VI method, which is based on forward KL and variational family parameterized by neural network. The analysis techniques are extended from widely studied NTK for two layer neural network. The authors also conducted experiments to verify the theoretical results.

**Strengths:**

The paper provided rigorous global convergence guarantee of amortized VI based on forward KL with exponential variational family and NTK. Although the results were asymptotic in the width of neural network, various numerical experiments were conducted to show that the theoretical results seem to hold with finite width.

**Weaknesses:**

The major concern is about novelty. What is the difference between the analysis in this paper and existing analysis of two layer neural network in supervised learning setting? Since the authors assumed that the variational family is exponential family, the optimization objective terms out to be a convex function of the output of two layer neural network, which has been widely studied. Although this is new in the literature of VI, the techniques used are not novel and can be easily extended from supervised learning setting to VI based on forward KL.

**Questions:**

The setting for variational inference in this paper assumes that data can be simulated during training (or with known latent variables), which is a bit different from standard VI where the data is fixed. How useful would it be for more general VI problems where the forward KL seems to be challenging to use?

**Limitations:**

The limitations are pointed out in the paper.

---

> ### Author Rebuttal · Authors · 2024-08-06
>
> Thank you very much for the review. Below, we try to answer your main questions and concerns.
>
> > *What is the difference between the analysis in this paper and existing analysis of two layer neural network in supervised learning setting...which has been widely studied?*
>
> Firstly, let us highlight that applying the NTK to the variational inference setting is novel and a contribution in its own right. Amortization (and the use of neural networks at all) is a relatively recent advancement in variational inference, and even today the main motivation for its use is for utility or cost-saving. An analysis of this type is novel in this setting and could have a significant impact towards adoption of amortization in VI. We have shown amortization is more than just a way to save compute, but actually has tangible benefits for optimization.
>
> Purely with respect to the NTK literature, our analysis is still novel and innovative in following ways:
> - We allow for network outputs $\eta := f(x; \phi)$ of arbitrary dimension ($\eta \in \mathbb{R}^q)$.
> - We study a general, convex loss function $\ell$.
> - We minimize the population loss $\mathbb{E}_{P(X)} \ell(X; f(X;\phi))$.
>
> Extending to the general setting above was necessary -- in the current literature, existing results could not be applied to analyze the expected forward KL objective.  Our generalizations contrast with the restrictive assumptions that are often featured in existing NTK analyses, which are often specifically focused on a particular setting consisting of i) a scalar-valued network and ii) mean-squared error loss. Additionally, virtually all works consider iii) empirical risk minimization, i.e., minimization of $\sum_{i=1}^n \ell(f(x_i;\phi), y_i)$ for finite training data, rather than for an infinite population, as we require.
>
>  To solve this problem, we have innovated on existing NTK results in the restrictive settings described above. One of our results that may be of particular interest to the NTK community is that of *uniform* convergence of the NTK to its limit, i.e. Proposition 2 and Proposition 3 in the appendices (l. 875 and 932, respectively). Previous works relied on pointwise convergence to analyze an empirical loss, but to analyze the population quantity $\mathbb{E}_{P(X)} \ell(X; f(X;\phi))$ we required uniform convergence of kernels and proved these results. Within our proofs, several techniques may also be of interest, in particular the utilization of generalized Gronwall inequalities (e.g., line 820) to bound differences over an interval $[0,T]$.
>
> Lemmas 4 and 5 (lines 900 and 921), proved using this approach, are (minor) standalone results that may aid additional work. Finally, we want to emphasize that although we write primarily for a Bayesian audience and restrict our analysis to the expected forward KL objective, our results apply to more general convex loss functions and may be extended the analysis of any objective that satisfies the convexity conditions, include those in settings beyond variational inference.
>
> > *The setting for variational inference in this paper assumes that data can be simulated during training...*
>
> We want to emphasize that the *expected* forward KL objective that we optimize is not challenging to use; ease of implementation of this method for practically any setting is one of the strengths of this approach. A forward KL divergence without the expectation over $P(X)$, on the other hand, is infeasible to optimize -- neither the objective nor its gradient are unbiasedly estimable, as estimating either quantity requires samples from the exact posterior, which cannot be obtained. We touched on this point briefly (line 104), but have now added exposition to contrast the differences between the forward KL and the expected forward KL for clarity.
>
> Our analysis of the *amortized* objective in this work may encourage more widespread use of the forward KL. The amortized objective resolves the point above: expectations over $P(X)$ and $P(\Theta \mid X)$ can be estimated unbiasedly as a single expectation over $P(\Theta, X)$ by using the ``trick'' of combining these as in Appendix B, and using ancestral sampling. The assumption we can simulate draws from $P(\Theta, X)$ is not restrictive; in fact, this assumption is strictly looser than assuming that the likelihood function $p(\theta, x)$ is readily available, a key assumption for any ELBO-based analysis. Modern probabilistic computing packages such as PyTorch and Pyro ensure that sampling from arbitrary distributions is straightforward via ancestral sampling of $\Theta \sim P(\Theta)$, $X \sim P(X \mid \Theta)$, etc. Expected forward KL minimization generalizes to standard VI settings with a single observable $x_{\textrm{true}}$ as well with minimal additional overhead -- we discussed such settings briefly in our submission (e.g. line 68, 99, and 354), but as also requested by other reviewers, we have added more substantial discussion on practical aspects to Section 6.
>
> A counterintuitive implication of our work is that simulation-based minimization of the expected forward KL may be useful even for non-amortized VI problems such as described above. Because simulation is often computationally inexpensive, and network training rapidly converges to a global minimizer, one can obtain a unique variational approximation $q(\theta; f(x_{\textrm{true}}, \phi))$. ELBO-based training, on the other hand, might yield vastly different posterior approximations to $p(\theta \mid x_{\textrm{true}})$ for different initializations or random seeds. Therefore, our work could simplify VI for practitioners. However, ELBO-based optimization still has many merits. We have added discussion of additional practical considerations to our discussion section, including cases where ELBO-based training may be preferable (e.g., for fitting model parameters alongside the variational approximation; if one wants to obtain mode-seeking approximations, etc.).

---

### Official Review · Reviewer_Dszp · 2024-07-13

**Soundness:** 3
**Presentation:** 3
**Contribution:** 3
**Rating:** 7
**Confidence:** 3

**Summary:**

The authors study convergence of forward-KL variational inference in the neural posterior estimation (NPE) setting with an exponential family variational distribution, whose natural parameters are produced by a neural network. For this setting, it is known that the forward-KL is convex in the natural parameters of the variational distribution. The authors show how this extends to the NPE setting, i.e. expected forward-KL, by linearity of the outer expectation and establish the existence of a global minimizer for the functional NPE objective. The authors further show that the limiting neural tangent kernel (NTK) of the network can be used to construct a Reproducing Kernel Hilbert space in which optimization via kernel gradient flow converges to the unique minimizer. Finally the authors show how this finding can be extended to the parametric setting by noting that, for certain network architectures, the parametric NTK tends to the limiting NTK as the width of the network tends to infinity. Interestingly, the authors demonstrate empirically that in practice infinite width might not be required to converge to a region close to the global minimizer.

**Strengths:**

The results are novel and theoretically interesting to the variational inference community. I found the paper, given its theoretical nature, well written and relatively easy to follow along.

**Weaknesses:**

The result stated in Lemma 1 is well known (unless I am missing something), please consider citing a textbook or relevant review paper for the fact that the forward-KL divergence is convex in the natural parameters of the variational distribution in the exponential family case.

The experiments are not directly compared to ELBO/IWBO based optimization, hence, it is unclear how big the actual gap in performance is. In settings where the true posterior is a member of the variational family, the minimizer of the forward-KL divergence also minimizes the reverse-KL and hence maximizes the ELBO and vice-versa. A plot that reports the symmetric-KL over training for both the ELBO and FKL objective would be insightful. In the end, if FKL-optimization indeed finds the minimizer (or close to it), the learned variational distribution should outperform ELBO/IWAE-optimization no matter which divergence (or corresponding bound) is used for comparison.

--- Edit ---

The authors addressed my concerns by adding appropriate references and providing additional results that demonstrate the utility of their approach. Consequently, I have raised my score to an Accept (7).

**Questions:**

-

**Limitations:**

-

---

> ### Author Rebuttal · Authors · 2024-08-06
>
> Thank you for the detailed review. We will incorporate your main suggestions as outlined below.
>
> > *The result stated in Lemma 1 is well known...please consider citing a textbook or relevant review paper*
>
> Although we agree that Lemma 1 follows easily from the standard properties of the exponential family, we have not been able to find the convexity of the forward KL in the natural parameter stated as such in any relevant paper, despite extensive searching. (We would welcome an exact citation if you are aware of one.) We suspect the result has not been derived before because the non-amortized objective and its gradient are not generally computable, rendering the convexity of the objective irrelevant. Similarly, the amortized objective is non-convex in the neural network parameters that are optimized, so thoughts of convexity have not been considered in-depth until now. Although the result is simple, it is fundamental to our analysis of the amortized problem.
>
> The best we can do with what we have found in a literature search is to add a reference to Proposition 2 of Wainwright \& Jordan's *Graphical Models, Exponential Families, and Variational Inference*, which proves that the log-partition function of an exponential family is convex in the natural parameter.
> Our lemma follows fairly easily from this proposition.
>
> > *...it is unclear how big the actual gap in performance is...a plot that reports the symmetric-KL over training for both the ELBO and FKL objective would be insightful.*
>
> This is a great suggestion; in the global response, we have provided an analysis of the quality of the solution found by minimizing the expected forward KL via several different objective measures. By all measures, we find that the minimizer of the expected forward KL outperforms the local optimum found by minimizing the negative ELBO -- the minimizer of the expected forward KL even has a lower reverse KL divergence to the exact posterior, despite not directly optimizing this quantity.

---

> > ### Comment · Reviewer_Dszp · 2024-08-12
> > **Re: Rebuttal by Authors**
> >
> > Thank you for your efforts in addressing my concerns!
> >
> > - After spending about 10 minutes searching for references, I indeed found it surprisingly difficult to find more specific ones. However, I believe Wainwright & Jordan is a solid reference.
> >
> > - I also think the additional results are excellent for completing the overall picture and make a compelling case for why the well-posedness of the underlying optimization problem might matter, even though it is often overlooked in common practice.
> >
> > Overall, after considering all reviews and corresponding rebuttals, I’ve decided to raise my score to an Accept (7). I believe this is a technically solid paper that presents novel theoretical results, rigorously extending existing NTK results and applying them to the VI setting. This makes it of interest to both the variational inference community and the broader NeurIPS audience!

---

### Author Rebuttal · Authors · 2024-08-06

We thank all the reviewers for their time and helpful comments. We are encouraged by your largely positive feedback and appreciate your thoughts on areas of improvement.

We have responded to each of your points individually in the review-specific rebuttals. In this global rebuttal, we wish to emphasize the significance of our contributions with respect to two research areas: variational inference and NTK analysis. We also provide additional experimental results (as suggested by Dszp, but likely of interest to all reviewers).

**A breakthrough in variational inference.**
In the VI community the use of posterior approximations parameterized by neural networks (i.e., amortization) is still a relatively recent phenomenon and is by no means ubiquitous. Our results are a major step towards expanding the use of amortization in variational inference.

We show that amortization has benefits beyond reducing computational costs; in fact, targeting the amortized objective can actually be
desirable because it admits a unique solution. Through our analysis, we resolve a significant complication for VI that previously seemed impenetrable: that of convergence only to a local optimum of the objective. The objective we consider in this work is convex, and we show that its gradient leads to the global optimum. Our results may help expand the use of VI in general. In practice, MCMC still remains more widespread than VI, in part due to potentially unreliable VI optimization -- our work is a significant step towards resolving this concern.

**A technically sophisticated adaptation of the NTK.**
 We emphasize that our results are not a mere application of existing NTK analyses to a different objective function. Existing NTK-based analyses (e.g., [1]-[5]) are restricted to specific settings that exclude variational inference. These restrictions typically include one or more of:
- a scalar-valued network output
- a mean-squared error loss
- a finite training set

These assumptions are simplifying and commonly used in practice for general machine learning, but do not apply in the settings commonly seen in variational inference, which may have i) parameters of arbitrary dimension, ii) diverse loss functions, and iii) population-type objective functions (i.e., an expectation over a continuous distribution). Bridging the gap between existing literature and analysis tailored to the VI objective we consider in this work required generalizing beyond the setting outlined above. To do so, we proved results for arbitrary loss functions and introduced new machinery (e.g., uniform rather than pointwise convergence of the NTK) necessary for establishing convergence over a continuous distribution of inputs.

[1] Generalization ability of wide residual networks, Lai et al.

[2] Gradient descent provably optimizes over-parameterized networks, Du et al.

[3] Linearized two-layer neural networks in high dimension, Ghorbani et al.

[4] Loss landscapes and optimization in over-parameterized non-linear systems and neural
    networks, Liu et al.

[5] On exact computation with an infinitely wide neural net, Arora et al.

**New experimental results**

Suggested by Dszp, we add a small case study that quantifies the *quality* of the variational approximation found by expected forward KL minimization -- in other words, how well it approximates the true posterior. We use a rotated MNIST example, with angle $\theta \sim \mathrm{Unif}[0, 2\pi]$ and for all $i=1, \dots 50$ we have $x_i \mid \theta \sim \mathcal{N}(\mathrm{Rotate}(\mu_i, \theta), \sigma^2)$, where $\mu_i$ is a fixed, synthetic MNIST digit, and $\mathrm{Rotate}(\cdot)$ applies a rotation of $\theta$ radians. The variational distribution is taken to be Gaussian with fixed $\sigma = 0.5$.

 We aim to measure the quality of the solutions found by ELBO-based optimization and expected forward KL minimization; because our approach tends toward the global optimum of the expected forward KL objective, we expect this solution to outperform local optima of other objectives by any metric used to measure the quality of the approximation.

In the attached pdf, we plot several quantities across fitting, where fitting either minimizes the negative of the ELBO or minimizes the expected forward KL. The negative ELBO objective was fit to a fixed dataset $x$ of $50$ images drawn from the model above with common rotation angle of 260 degrees. We denote these images as $x_{\textrm{true}}$, and the true rotation angle $\theta_{\textrm{true}}$. The variational approximation is denoted as $q(\theta \mid x_{\textrm{true}})$.

For the negative ELBO objective, the estimated angle converges to about 90 degrees, while expected forward KL minimization (as expected) is centered near correct angle value (Figure 1). This translates to a better held-out negative log likelihood on the true latent angle value, as expected (Figure 2). We also display both the forward (Figure 3) and reverse (Figure 4) KL divergences across training. These quantities are difficult to estimate exactly -- we compute the forward KL using importance sampling with the prior $p(\theta)$ as the proposal and $K=1000$ importance samples, and the reverse KL is approximated using the ELBO plus an estimate of the log evidence.

Perhaps surprisingly, expected forward KL minimization outperforms optimizing the negative ELBO *even with respect to the negative ELBO objective function*. In other words, the variational distribution fit to minimize the expected forward KL turns out to have a lower (better) reverse KL value that the distribution fit to minimize reverse KL. This arises from the intuition that a global optimum of a different objective may be preferable to a local optimum of one's original target. This is the motivation behind our manuscript: the global minimum of the expected forward KL is closer to the exact posterior than a local optimum of the ELBO.

---

### Decision · Program_Chairs · 2024-09-25

**Decision:**

Accept (poster)

**Comment:**

This paper examines the variational inference problem and, unlike previous works, provides a global convergence guarantee. The analysis focuses on forward KL-based variational inference, where the variational family is parameterized by neural networks. The authors adapt NTK-based analysis for two-layer neural networks to achieve the desired results for their specific variational inference problem.

Almost all reviewers agree that the paper merits acceptance, and after carefully reading the rebuttal and discussion, I tend to concur. Please incorporate the reviewers’ feedback into the final version of the paper, with particular attention to the following concerns:


1- Include the empirical comparisons to the ELBO/IWAE methods (Response to  Rev DszP).

2- Include the ref for Lemma 1 and clarify its efficiency. (Response to Rev DszP and grF1)

3-  Include a discussion about when your method is more efficient than ELBO-based methods. (Response to Rev gbq5)